# Adapting Fairness Interventions to Missing Values

**Raymond Feng**[1]**, Flavio P. Calmon**[1]**, Hao Wang**[2]
[1]John A. Paulson School of Engineering and Applied Sciences, Harvard University
[2]MIT-IBM Watson AI Lab
raymond_feng@college.harvard.edu, flavio@seas.harvard.edu, hao@ibm.com

## Abstract

Missing values in real-world data pose a significant and unique challenge to algorithmic fairness. Different demographic groups may be unequally affected by missing data, and the standard procedure for handling missing values where first data is imputed, then the imputed data is used for classification—a procedure referred to as "impute-then-classify"—can exacerbate discrimination. In this paper, we analyze how missing values affect algorithmic fairness. We first prove that training a classifier from imputed data can worsen the achievable values of group fairness and average accuracy. This is because imputing data results in the loss of the missing pattern of the data, which often conveys information about the predictive label. We present scalable and adaptive algorithms for fair classification with missing values. These algorithms can be combined with any preexisting fairness-intervention algorithm to handle missing data while preserving information encoded within the missing patterns. Numerical experiments with state-of-the-art fairness interventions demonstrate that our adaptive algorithms consistently achieve higher fairness and accuracy than impute-then-classify across different datasets.

## 1 Introduction

Missing values exist in many datasets across a wide range of real-world contexts. Sources of missing values include underreporting in medical information [18], structural missingness from data collection setups [43], and human error. Missing data poses a challenge to algorithmic fairness because the mechanisms behind missing values in a dataset can depend on demographic attributes such as race, sex, and socioeconomic class. For example, in housing data, non-white housing applicants or sellers may be less likely to report their race due to concerns of discrimination [51]. Real-world data collection and aggregation processes can also lead to disparate missingness between population groups. If missing data is not handled carefully, there is a high risk of discrimination between demographic groups with different rates or patterns of missing data. For example, low-income patients can have more missing values in medical data involving the results of costly medical tests [37], making them less likely to qualify for early interventions or get treatment for insufficiently severe conditions [17].

The issue of missing values has not been sufficiently addressed in current fairness literature. Most fairness-intervention algorithms that have been proposed are designed for complete data and cannot be directly applied to data with missing values [e.g., 1, 2, 7, 10, 25, 32, 60, 61]. A common approach to handle missing values is to first perform imputation, filling in missing entries with appropriate values, and then apply an off-the-shelf fairness-intervention algorithm to the imputed data (*impute-then-classify*). However, this standard procedure has several limitations. In many contexts and data formats, imputation is not an appropriate method for handling missing values. In healthcare data, some missing data is categorized as "intended missing data," which, for example, the National Cancer Institute does not recommend imputing [43]. For instance, if a respondent answers NO to the question "Were you seen for an illness or injury?", they should *skip* the question "How often

did you get care for an illness or injury as soon as you wanted?". Imputing the second question does not make sense knowing the format of the data. The missingness pattern of the data can also convey important information about the predictive label. For instance, in surveys with questions about alcohol consumption history, missing responses are often correlated with alcoholism due to pervasive social stigma surrounding alcoholism [39]. A model trained on imputed data will suffer from reduced performance, especially for groups with higher missing data rates.

In this work, we investigate how missing values in data affect algorithmic fairness. We focus on the setting where input features of predictive models may have missing values. Although a growing body of research [27, 62] is dedicated to addressing the issue of missing group (sensitive) attributes, the problem of missing input features is equally crucial and widespread yet less studied. Our objective in this paper is to bridge this gap and offer a comprehensive study encompassing theory and algorithms for training fair classifiers in the presence of missing input features, particularly when missingness patterns vary per population group (as observed in practice).

First, we show that the performance of a fair classifier trained with impute-then-classify can be significantly lower than one trained directly from data with missing values. Impute-then-classify may fail because imputation loses information from the missing pattern of the data that can be useful for prediction. Our result is information-theoretic: a data scientist cannot improve this fundamental limit by altering the hypothesis class, using a different imputation method, or increasing the amount of training data.

Next, we introduce methods for adapting fairness-intervention algorithms to missing data. In Section 4, we present three adaptive algorithms for training fair linear classifiers on data with missing values, and in Section 5, we introduce an algorithm that adapts general fairness interventions to handle missing values. We treat the linear case separately because the interpretability of linear classifiers facilitates their widespread use in practice [11, 47, 54, 56]. Our algorithms work by modifying the dataset to preserve the information encoded in the missing patterns, then applying an off-the-shelf fairness-intervention algorithm to the new dataset. This avoids the information-theoretic limitation of impute-then-classify described in our theoretical result above. Most importantly, our algorithms are flexible because they can be combined with any preexisting fairness intervention to handle missing values fairly, accurately, and efficiently. Other advantages include scalability to high-dimensional data, generalizability to new data, and the ability to handle new missing patterns that occur only at testing time.

We benchmark our adaptive algorithms through comprehensive numerical experiments. We couple the adaptive algorithms with state-of-the-art fair classification algorithms and compare the performance of these methods against impute-then-classify. Our results show that the adaptive algorithms achieve favorable performance compared to impute-then-classify on several datasets. The gap in performance is more pronounced when missing data patterns encode more information, such as when data missingness is correlated with other features, i.e., not missing uniformly at random. Although the relative performance of the adaptive methods depends on the data distribution, even a simple method such as adding missing indicators as input features consistently achieves better fairness-accuracy curves than impute-then-classify. When applying fairness interventions on missing data, we find that utilizing information from missing patterns is crucial to mitigate the fairness and accuracy reduction incurred by imputation. Our results suggest that practitioners using fairness interventions should preserve information about missing values to achieve the best possible group fairness and accuracy.

## 1.1 Related work

**Missing values in supervised learning.** A line of recent works has investigated the limitations of performing imputation prior to classification [4, 26, 36]. In a remarkable paper, Le Morvan et al. [36] found that even if an imputation method is Bayes optimal, downstream learning tasks generally require learning a discontinuous function, which weakens error guarantees. This motivated a neural network algorithm that jointly optimizes imputation and regression using an architecture for learning optimal predictors with missing values introduced in [35, 42]. Among related work in supervised learning with missing values, the closest to ours is [4] and [31]. Bertsimas et al. [4] devised several algorithms for a general linear regression task with missing values where the missingness pattern of the data is utilized to make predictions. In particular, the static regression with affine intercept algorithm involves adding missing indicators as feature variables and has been found to improve performance for standard prediction tasks [45, 50, 52, 57]. Our work generalizes their algorithms

to fair classification tasks. We also extend the family of ensemble approaches in Khan et al. [31] to fair classification using a general, non-linear base classifier. Our approach is similar to [49] in using bootstrapping to satisfy strengthened fairness conditions; a key difference is that we explicitly use (known) sensitive attributes and labels when drawing subsamples to ensure sample diversity.

**Missing values in fair machine learning.** Discrimination risks related to missing values were studied in [6, 14, 23, 48, 53, 58]. Caton et al. [6], Fernando et al. [14], Jeong et al. [23], Mansoor et al. [40], Schelter et al. [48], Zhang and Long [63] provided empirical assessments on how different imputation algorithms impact the fairness risks in downstream prediction tasks. These studies have highlighted that the choice of imputation method impacts the fairness of the resulting model and that the relative performance of different imputation methods can vary depending on the base model. To mitigate discrimination stemming from missing values, researchers have proposed different methods for fair learning with missing values for tabular data [23, 58] and for graph data [53]. Among these works, the closest one to ours is Jeong et al. [23] which proposed a fair decision tree model that uses the MIA approach to obtain fair predictions without the need for imputation. However, their algorithm hinges on solving a mixed integer program, which can be computationally expensive.

Another related line of work studies how one can quantify the fairness of a ML model when the required demographic information (i.e., sensitive group attributes) is missing. Methods to estimate fairness of models in the absence of sensitive group attributes have been developed using only observed features in the data [27] or complete case data [62]. Multiple works have also studied fairness when predictive labels themselves are missing. Wu and He [59] propose a model-agnostic approach for fair positive and unlabeled learning. Ji et al. [24] propose a Bayesian inference approach to evaluate fairness when the majority of data are unlabeled. Lakkaraju et al. [33] propose an evaluation metric in settings where human decision-makers selectively label data and make decisions using unobserved features. In contrast, we focus on algorithm design for the case of missing non-group attribute input features.

## 2 Preliminaries

Throughout, random variables are denoted by capital letters (e.g., $X$), and sets are denoted by calligraphic letters (e.g., $\mathcal{X}$). We consider classification tasks where the goal is to train a classifier $h$ that uses an input feature vector $\boldsymbol{x} = (x_1, \cdots, x_d)$ to predict a label $y$. We restrict our domain to binary classification tasks where $y \in \{0, 1\}$, but our analysis can be directly extended to multi-class tasks. In practice, $\boldsymbol{x}$ may contain missing values, so we let $\boldsymbol{x} \in \prod_{i=1}^{d} (\mathcal{X}_i \cup \{\text{NA}\})$.

We adopt the taxonomy in [38] and consider three main mechanisms of missing data: missing completely at random (MCAR), where the missingness is independent of both the observed and unobserved values, missing at random (MAR), where the missingness depends on the observed values only, and missing not at random (MNAR), where the missingness depends on the unobserved values. We introduce a missingness indicator vector $\boldsymbol{m} \in \mathcal{M} = \{0, 1\}^d$ that represents the missingness pattern of $\boldsymbol{x}$: the missingness indicator $m_j = 1$ if and only if the $j$th feature of $\boldsymbol{x}$ is missing.

We evaluate differences in classifier performance with respect to a group (sensitive) attribute $S \in \mathcal{S}$. Fairness-intervention algorithms often aim to achieve an optimal fairness-accuracy trade-off by solving a constrained optimization problem

$$\max_h \mathbb{E}\left[\mathbb{I}(h(X) = Y)\right] \quad \text{s.t. } \mathsf{Disc}(h) \leq \epsilon \tag{1}$$

for some tolerance threshold $\epsilon \geq 0$. Here $\mathbb{I}$ is the indicator function, which equals 1 when the classifier $h$ produces the correct prediction $h(X) = Y$, and otherwise, it equals 0. $\mathsf{Disc}(h)$ represents the violation of a (group) fairness constraint. For example, we can define $\mathsf{Disc}(h)$ for equalized odds as

$$\mathsf{Disc}(h) = \max_{y, \hat{y}, s, s'} |\Pr(h(X) = \hat{y}|Y = y, S = s) - \Pr(h(X) = \hat{y}|Y = y, S = s')|.$$

Equalized odds addresses discrimination by constraining the disparities in false positive and false negative rates between different sensitive groups. Analogous expressions exist for other group fairness constraints such as equality of opportunity and statistical parity [13, 20]. In the case where multiple, overlapping sensitive attributes exist in the data, our framework can be extended to the multiaccuracy and multicalibration fairness measures introduced in Kim et al. [32] and Hébert-Johnson et al. [21].

## 3 Characterization of Fairness Risk on Imputed Data

Methods for learning a fair classifier can fail when data contains missing values. For example, many fairness interventions use base classifiers such as linear classifiers and random forest that do not support missing values [e.g., 2, 61]. Others use gradient-based optimization methods that require a continuous input space [e.g., 1]. Impute-then-classify is the most common way of circumventing this issue. However, a key limitation of impute-then-classify is that imputing missing values loses information that is encoded in the missingness of the data. If the data contains missing values that are not MCAR, the fact that a feature is missing can be informative. A model trained on imputed data loses the ability to harness missingness to inform its predictions of the label. We introduce a theoretical result that rigorously captures this intuition.

We represent an imputation mechanism formally as a mapping $f_{\mathrm{imp}} : \prod_{i=1}^{d} (\mathcal{X}_i \cup \{\mathtt{NA}\}) \to \prod_{i=1}^{d} \mathcal{X}_i$ and denote the imputed features by $\hat{\boldsymbol{x}} = f_{\mathrm{imp}}(\boldsymbol{x})$. We define the underlying data distribution by $P_{S,X,Y}$ and the distribution of imputed data by $P_{S,\hat{X},Y}$. We use the mutual information to measure the dependence between the missing patterns $M$ and the label $Y$:

$$I(M;Y) \triangleq \sum_{\boldsymbol{m} \in \mathcal{M}} \sum_{y \in \{0,1\}} P_{M,Y}(\boldsymbol{m},y) \log \frac{P_{M,Y}(\boldsymbol{m},y)}{P_M(\boldsymbol{m})P_Y(y)}.$$

Recall the constrained optimization problem in (1) for training a fair classifier. When the optimization is over *all* binary mappings $h$, the optimal solution only depends on the data distribution $P_{S,X,Y}$ and the fairness threshold $\epsilon$. In this case, we denote the optimal solution of (1) by $F_\epsilon(P_{S,X,Y})$. The next theorem states that even in the simple case where a single feature has missing values, impute-then-classify can significantly reduce a classifier's achievable performance under a group fairness constraint when the missingness pattern of the data is informative with respect to the predictive label.

**Theorem 1.** *Suppose that $X$ is composed by a single discrete feature (i.e., $d = 1$) and $\mathrm{Disc}(h)$ is the equalized odds constraint. Let $g : [0,1] \to \mathbb{R}$ be the binary entropy function: $g(a) = -a \log(a) - (1-a) \log(1-a)$. For any $\epsilon$ and $a < 1/3$, there exists a data distribution $P_{S,X,Y}$ such that $I(M;Y) = g(a)$ and*

$$\sup_{f_{\mathit{imp}}} F_\epsilon(P_{S,\hat{X},Y}) \leq F_\epsilon(P_{S,X,Y}) - a.$$

This information-theoretic result shows that applying a fairness-intervention algorithm to imputed data can suffer from a significant accuracy reduction which grows with the mutual information between the missing patterns and the label. Note that the supremum is taken over *all* imputation mechanisms; the inequality holds regardless of the type of classifier, imputation method used, or amount of training data available. In other words, the negative impact to performance due to information loss from imputation is unavoidable if using impute-then-classify.

## 4 Adapting Linear Fair Classifiers to Missing Values

In the last section, we showed how a classifier trained from impute-then-classify may exhibit suboptimal performance under a (group) fairness constraint. We now address this issue by presenting adaptive algorithms that can be combined with fairness interventions to make fair and accurate predictions on data with missing values. We extend prior work in [4] to a fair classification setting.

We focus first on adapting fairness interventions to missing values in linear classifiers, and present results for non-linear methods in the next section. We treat the linear case separately since, despite their simplicity, linear classifiers are widely used in practice and in high-stakes applications (e.g., medicine, finance) due to their interpretability [11, 47, 54, 56]. The adaptive algorithms we propose allow a model to handle any missing pattern that appears at training and/or test (deployment) time.

We propose three methods tailored to linear classification: (i) adding missing values as additional features, (ii) affinely adaptive classification, and (iii) missing pattern clustering. Our methods encode missing value patterns via adding new features in the case of (i) and (ii) and partitioning the dataset in the case of (iii), allowing a fairness intervention to then be applied on the encoded data. For example, if the fairness intervention is a post-processing scheme, a linear classifier is fit on the encoded data and then modified by the intervention. For in-processing schemes [1, 7, 30], the fair linear classifier is simply trained on the encoded data. The three methods are illustrated in Figure 4a (Appendix C).

## 4.1 Adding missing indicator variables

Our first adaptive algorithm is to pass $\boldsymbol{m}$ as part of the input feature vector and impute missing values by zero. Letting $\hat{\boldsymbol{x}}$ denote the imputed original features, our new feature vectors are now of the form $(\hat{\boldsymbol{x}}, m) \in (\prod_{i=1}^d \mathcal{X}_i) \times \{0, 1\}^d$, and we can then train an off-the-shelf fairness-intervention algorithm on the new data. This approach is highly scalable as the number of features in the new data is $2d$. We can interpret learning a linear classifier $\boldsymbol{w} = (w_1, \ldots, w_d, b_1, \ldots, b_d)$ on the transformed dataset (where $b_j$ is the coefficient of $m_j$) as equivalent to jointly learning a linear classifier $\boldsymbol{w}' = (w_1, \ldots, w_d)$ on the original data and an imputation method that replaces missing values in feature $j$ with $\frac{b_j}{w_j}$ [4]. Alternatively, noting that $\boldsymbol{w}^\top(\hat{x}, m) = \sum_{j=1}^d w_j x_j (1 - m_j) + \sum_{j=1}^d b_j m_j$, adding missing indicators effectively allows the bias term to be adjusted by $b_j$ when feature $j$ is missing.

Intuitively, using missing indicators captures the information encoded in the missingness patterns of the data. We provide an information-theoretic interpretation of this intuition in Appendix B.

## 4.2 Affinely adaptive classification

As mentioned above, adding missing indicator variables allows the bias term of a linear classifier to be adjusted by a constant when a feature is missing. We can treat the coefficients of the classifier in a similar way. Affinely adaptive regression includes additional input features of the form $m_k(1 - m_j)x_j$ for $j, k \in [d]$ with $j \neq k$, then applying zero imputation and an off-the-shelf fairness intervention. This can be interpreted as training a linear classifier with coefficients $\boldsymbol{w}_0 + \mathbf{W}\boldsymbol{m}$, where $\boldsymbol{w}_0$ is an initial set of coefficients and $W_{jk}$ denotes the adjustment to $w_j$ when $x_k$ is missing:

$$(\boldsymbol{w}_0 + \mathbf{W}\boldsymbol{m})^\top \hat{\boldsymbol{x}} = \sum_{j=1}^d w_{0j}(1 - m_j)x_j + \sum_{j=1}^d \sum_{k=1}^d W_{jk} m_k (1 - m_j) x_j.$$

Affinely adaptive classification generalizes simply adding missing indicators by including a bias term as an additional input feature. However, a significant drawback is that affinely adaptive classification requires $O(n_{\text{miss}}d)$ features where $n_{\text{miss}}$ is the number of original features that contain missing values, though this number may be small in practice.

## 4.3 Missing pattern clustering

The previous two algorithms transform the original dataset by adding additional input features encoding data missingness. We now introduce a two-step missing pattern clustering algorithm. First, the missing patterns in the data are split into clusters. Then, an off-the-shelf fairness intervention algorithm is applied on each cluster separately. Rather than encoding data missingness with additional input features, this algorithm avoids the limitation of impute-then-classify by using missing patterns to partition the dataset. We also impose group fairness constraints in the clustering process to ensure sufficient sample size and diversity in each cluster. This allows the fair classifiers on each cluster to generalize to fresh data.

During clustering, we split all possible missing patterns $\mathcal{M}$ into $Q$ clusters $\{\mathcal{M}_q\}_{q=1}^Q$ using a recursive partitioning method. Let $\mathcal{P}$ denote the current (intermediate) partition, so initially $\mathcal{P} = \{\mathcal{M}\}$. At each step, we split each cluster $\mathcal{M}_q \in \mathcal{P}$ based on the missingness of some feature $j$ into two new sets $\mathcal{M}_q^{j0} = \{\boldsymbol{m} \in \mathcal{M}_q : m_j = 0\}$ and $\mathcal{M}_q^{j1} = \{\boldsymbol{m} \in \mathcal{M}_q : m_j = 1\}$. We split the training dataset correspondingly into $\mathcal{I}_q^{j0} = \{(\boldsymbol{x}_i, y_i) : \boldsymbol{m}_i \in \mathcal{M}_q^{j0}\}$ and $\mathcal{I}_q^{j1} = \{(\boldsymbol{x}_i, y_i) : \boldsymbol{m}_i \in \mathcal{M}_q^{j1}\}$. To determine the feature on which to split, we introduce a loss function:

$$L(\mathcal{M}_q, j) \triangleq \min_{\boldsymbol{w}} \sum_{i \in \mathcal{I}_q^{j0}} \ell(y_i, \boldsymbol{w}^\top \boldsymbol{x}_i') + \min_{\boldsymbol{w}} \sum_{i \in \mathcal{I}_q^{j1}} \ell(y_i, \boldsymbol{w}^\top \boldsymbol{x}_i')$$

where $\boldsymbol{x}_i'$ is $\boldsymbol{x}_i$ with missing values imputed by zero. In other words, $L(\mathcal{M}_q, j)$ is the total loss from training linear classifiers on $\mathcal{I}_q^{j0}$ and $\mathcal{I}_q^{j1}$.

$L(\mathcal{M}_q, j)$ satisfies two desirable properties. First, it can be calculated efficiently as it does not include fairness risk, which is crucial as there are $O(d)$ candidate features for each split. Second, $L(\mathcal{M}_q, j)$ guarantees that the total loss with respect to the standard classifiers

$$\sum_{\mathcal{M}_q \in \mathcal{P}} \min_{\boldsymbol{w}} \sum_{i \in \mathcal{I}_q} \ell(y_i, \boldsymbol{w}^\top \boldsymbol{x}_i')$$

never increases after a split, provided that we split a cluster $\mathcal{M}_q$ only if $L(\mathcal{M}_q, j) < \min_{\boldsymbol{w}} \sum_{i \in \mathcal{I}_q} \ell(y_i, \boldsymbol{w}^\top \boldsymbol{x}'_i)$, i.e. the two new classifiers collectively perform at least as well as a classifier trained on the whole cluster.

To address the fairness concerns of having clusters with small sample size or sample imbalance, we require each cluster to have balanced data from each group. A feature $j$ is considered for splitting only if $\mathcal{I}_q^{j0}$ and $\mathcal{I}_q^{j1}$ satisfy bounded representation for chosen parameters $\alpha, \beta$:

$$\beta \leq \frac{|\{\boldsymbol{x}_i \in \mathcal{I}_q^{jk} | s_i = s\}|}{|\mathcal{I}_q^{jk}|} \leq \alpha, \quad s \in \mathcal{S}, k \in \{0, 1\}.$$

We also impose a minimum permissible cluster size $k$ for $\mathcal{I}_q^{j0}$ and $\mathcal{I}_q^{j1}$. We will denote the set of features $j$ that satisfy these constraints by $G(\mathcal{M}_q)$. At each splitting iteration for each cluster $\mathcal{M}_q$, we split on $j^* \in \arg \min_{G(\mathcal{M}_q)} L(\mathcal{M}_q, j)$. After obtaining clusters $\{\mathcal{M}_q\}_{q=1}^Q$, we apply an off-the-shelf fairness intervention to train a fair linear classifier on each cluster. The fairness constraints we enforce during clustering ensure that the classifiers for each cluster can generalize to fresh data. We summarize our recursive partitioning method in Algorithm 1 (Appendix C).

## 5    Adapting Non-Linear Fair Classifiers to Missing Values

The algorithms above take advantage of the interpretability of linear classifiers to "deactivate" missing features via zero imputation. We now provide a general algorithm for nonlinear classifiers where the effect of zero imputation may be less clear. We introduce `FairMissBag`, a fair bagging algorithm that can be applied to *any* fair classifier to handle missing values. The algorithm proceeds in three steps (see Figure 4b, Appendix C for an illustration).

- The training data $\mathcal{D}$ is separated into groups by sensitive attribute and label $\mathcal{D}_{s,y} = \{(\boldsymbol{x}_i, y_i, s_i) | s_i = s, y_i = y\}$. From each $\mathcal{D}_{s,y}$, we sample $|\mathcal{D}_{s,y}|$ data points uniformly with replacement. We combine these samples to create a new dataset of the same size as the original dataset. Missing indicators are added to each data point and an imputation method is used on each subsample separately. We repeat this process $B$ times, creating $B$ distinct datasets.

- Apply an off-the-shelf fairness intervention to train a classifier on each resampled dataset, yielding $B$ classifiers in total.

- To make a prediction on a new data point, one of the $B$ fair classifiers is chosen uniformly at random. We return the prediction from this classifier. Alternatively, if the classifiers output a probability or score for each label, the scores from all classifiers are averaged to generate an output.

In the first step, the uniform resampling procedure ensures sufficient sample size and diversity by preserving the balance of data across sensitive attribute and outcome, relative to the original data. Additionally, the inclusion of missing indicators as input features (Section 4.1) prevents the issue of reduced fairness-accuracy curve described in Theorem 1. Our procedure is similar to the fair resampling procedure introduced in Kamiran and Calders [28] and Wang and Singh [58] except that we relax the independence assumption of $S$ and $Y$. This is advantageous in, for example, healthcare scenarios where a demographic group may have a higher prevalence of a disease (e.g., determined by age). It is likely desirable for a model to select patients from this group for treatment at a higher rate rather than enforce independence between $S$ and $Y$ by equalizing the selection rate between groups.

We show that our ensemble strategy in step 3 also preserves fairness by extending the analysis in [19]. We use equalized odds here as an example; the same fairness guarantee also holds for other fairness constraints such as statistical parity and equality of opportunity. If the base classifiers output scores (probabilities) for each class, an analogous argument justifies averaging the scores.

**Lemma 1.** *Suppose we have an ensemble of classifiers $\mathcal{C} = \{\mathcal{C}_j\}_{j=1}^M$ such that each classifier satisfies an equalized odds constraint, i.e. there exists $\varepsilon$ such that for all $j$,*

$$|\Pr(\mathcal{C}_j(X) = 1 | Y = y, S = s) - \Pr(\mathcal{C}_j(X) = 1 | Y = y, S = s')| \leq \varepsilon \quad \forall y \in \{0, 1\}, \ s, s' \in \mathcal{S}.$$

*Let $p(j)$ denote a probability distribution over the classifiers and $\mathcal{C}(\boldsymbol{x})$ be the prediction obtained by randomly selecting a base classifier $\mathcal{C}_k$ from $p(j)$ and computing $\mathcal{C}_k(\boldsymbol{x})$. Then $\mathcal{C}$ also satisfies the equalized odds constraint.*

# 6 Numerical Experiments

We evaluate our adaptive algorithms with a comprehensive set of numerical experiments by comparing our algorithms coupled with state-of-the-art fairness intervention algorithms on several datasets. We describe the setup and implementation and discuss the results of the experiments. Refer to Appendix D for additional experiment details.

## 6.1 Experimental setup

**Datasets.** We test our adaptive algorithms on COMPAS [34], Adult [12], the IPUMS Adult reconstruction [9, 15], and the High School Longitudinal Study (HSLS) dataset [22]. We refer the reader to Appendix D.2 for details on pre-processing each dataset and generating missing values for COMPAS and the two versions of Adult. The sensitive attribute is race in COMPAS and sex in Adult.

HSLS is a dataset of over 23,000 high school students followed from 9th through 12th grade that includes surveys from students, teachers, and school staff, demographic information, and academic assessment results [22]. We use an 11-feature subset of the dataset. The sensitive attribute is race (White/Asian versus Black, Hispanic, Native American, and Pacific Islander) and the predictive label is a student's 9th grade math test score. We consider only data points where both race and 9th grade math test score are present because our experiments use fairness interventions that use knowledge of the group attribute and label to calculate fairness metrics. HSLS exhibits high rates of missingness as well as significant disparities in missingness with respect to race. For instance, 35.5% of White and Asian students did not report their secondary caregiver's highest level of education; this proportion increases to 51.0% for underrepresented minority students [23].

**Fairness intervention algorithms.** We use `DispMistreatment` [61] and `FairProjection` [2] as two benchmark algorithms and defer additional experimental results using other benchmarks (`Reduction` [1], `EqOdds` [20], `ROC` [29], `Leveraging` [8]) to Appendix E. All of these algorithms are designed for complete data and require missing values to be imputed before training.

For each combination of adaptive algorithm and fairness intervention, we plot the fairness-accuracy tradeoff curve obtained from varying hyperparameters (see Appendix D.3). We obtain a convex curve by omitting points that are Pareto dominated, i.e. there exists another point with higher accuracy and lower discrimination. The fairness metrics used are FNR difference and mean equalized odds, defined as $\frac{1}{2}$ (FNR difference + FPR difference).

## 6.2 Linear fair classifiers

We test different combinations of the above fairness interventions with the adaptive algorithms in Section 4 using logistic regression as the base classifier. For the missing pattern clustering algorithm, we reserve part of the training set as validation to calculate the clustering objective function. We present results for `FairProjection` on IPUMS Adult/COMPAS and `FairProjection` and `DispMistreatment` on HSLS. We defer results for other benchmarks to Appendix E.

**IPUMS Adult/COMPAS.** Figure 1 displays fairness-accuracy curves for COMPAS and the IPUMS Adult reconstruction. We observe that for the datasets with MCAR and MAR missing values, the three methods tested have comparable performance with impute-then-classify. However, for the MNAR datasets, adding missing indicators and affinely adaptive classification significantly outperform the baseline especially with regard to accuracy, which is consistent with Theorem 1. We omit results for missing pattern clustering as the majority of trials did not find multiple clusters.

**HSLS.** Figure 2 displays the fairness-accuracy curves for `DispMistreatment` and `FairProjection` on HSLS. For `DispMistreatment`, adding missing indicators achieves marginally improved performance over the baseline, while affinely adaptive classification achieves the highest accuracy but at the cost of fairness. For `FairProjection`, all three adaptive methods outperform the baseline.

**Discussion.** The experiments on linear fair classifiers show that using the adaptive fairness-intervention algorithms proposed in Section 4 consistently outperform impute-then-classify in terms of accuracy and fairness across different datasets. The performance improvement of the adaptive

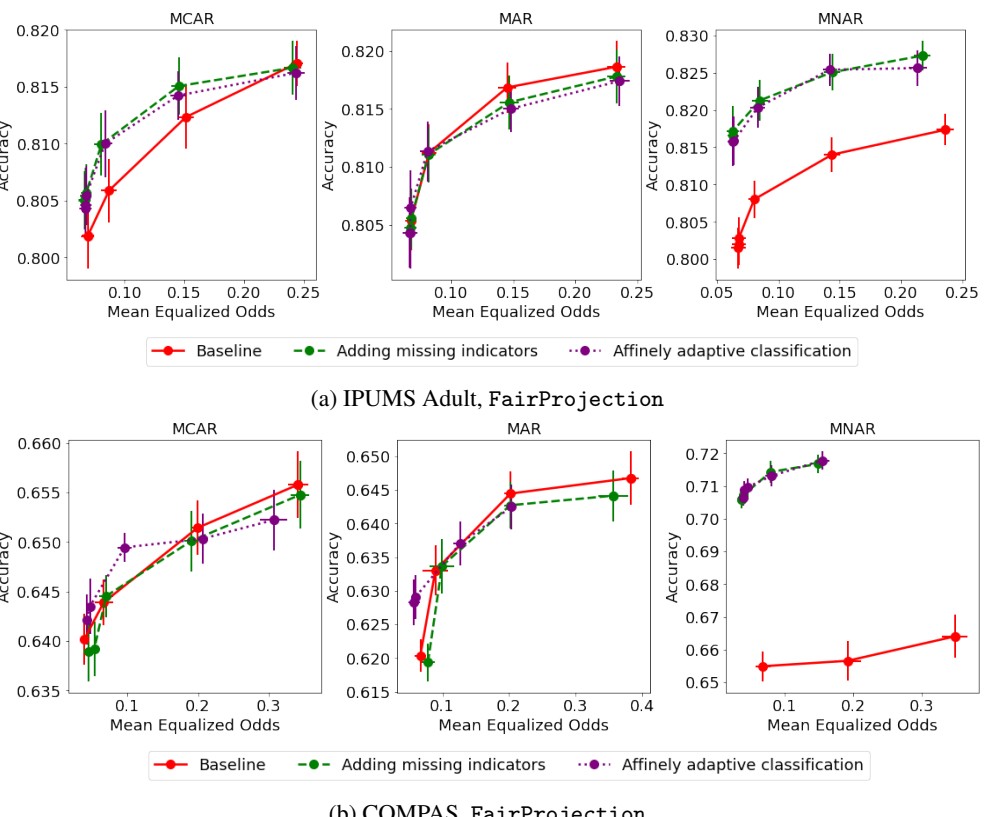

(a) IPUMS Adult, `FairProjection`

(b) COMPAS, `FairProjection`

Figure 1: Comparison of adaptive fairness-intervention algorithms on IPUMS Adult and COMPAS using `FairProjection`. For each original dataset, we generate three datasets with missing values corresponding to MCAR, MAR, and MNAR mechanisms (Appendix D.2). Error bars depict the standard error of 10 runs with different train-test splits.

fairness-intervention algorithms also increases when the missing patterns of the data encode significant information about the label, as is the case in the MNAR COMPAS/Adult datasets. Interestingly, although affinely adaptive classification and missing pattern clustering are both more expressive variants of adding missing indicators, adding missing indicators was able to achieve comparable performance to both methods in most cases. We include an additional experiment in Appendix E on a synthetic dataset (see Appendix D.1) where missing pattern clustering outperforms all other adaptive algorithms.

In short, we believe that the best adaptive algorithm is not universal. Our suggestion is to consider the choice of algorithm as a hyperparameter and select by using a validation set and considering the trade-offs between the methods proposed. Missing value indicators, though performing worse on average, has the benefit of being simple and interpretable. Affinely adaptive classification provides a potential accuracy gain at the cost of additional complexity, and missing pattern clustering should be preferred when there is sufficient data to cluster missing patterns in an advantageous way. Regardless of this choice, however, our results show that preserving information about missing values is important to achieve the best possible group fairness and accuracy.

### 6.3 Non-linear fair classifiers

We test the `FairMissBag` ensemble method proposed in Section 5 with `FairProjection`, `Reduction`, and `EqOdds`. The base classifier for all fairness-intervention algorithms is a random forest consisting of 20 decision trees with maximum depth 3. We run experiments with three imputation methods: mean, K-nearest neighbor (KNN), and iterative imputation [44]. We present results for `FairMissBag` on HSLS and defer additional experimental results to the appendix.

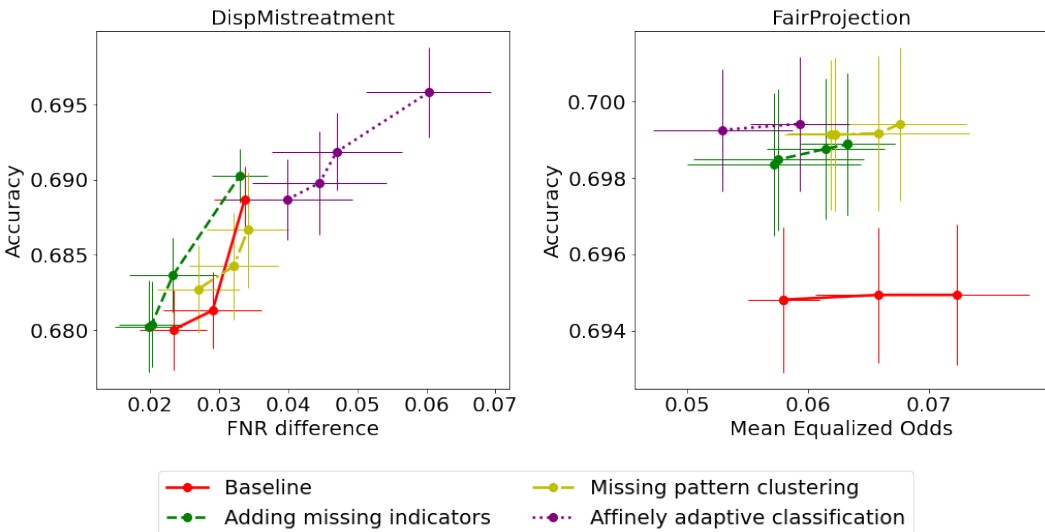

Figure 2: Comparison of adaptive fairness-intervention algorithms on HSLS dataset, using `DispMistreatment` (left) and `FairProjection` (right). Error bars depict the standard error of 10 runs with different train-test splits.

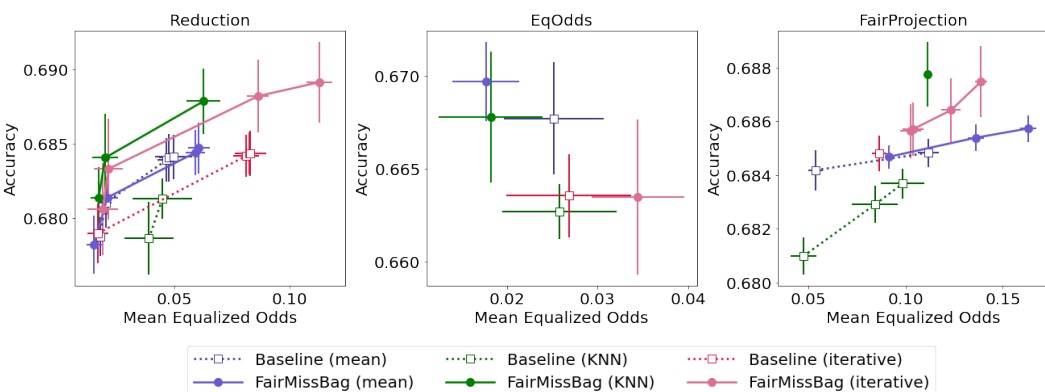

Figure 3: Results for `FairMissBag` on HSLS, using `Reduction`, `EqOdds`, and `FairProjection`. Fairness-accuracy curves under mean, KNN, and iterative imputation are shown, respectively. Error bars depict the standard error of 5 runs with different train-test splits.

Figure 3 displays the results of the `FairMissBag` experiment on HSLS. We observe that for `Reduction`, `FairMissBag` yields a meaningful fairness-accuracy improvement for KNN and iterative imputation. A similar improvement is observed for `EqOdds` for mean and KNN imputation. For `FairProjection`, `FairMissBag` generally extends the fairness-accuracy curve, providing higher accuracy with a tradeoff of increased fairness risk.

**Discussion.** We observe that using `FairMissBag` yields fairness and accuracy improvements over impute-then-classify. Additionally, for both accuracy and fairness, the relative performance of the three imputation methods we tested depends on the dataset used and the fairness-intervention algorithm applied. We conclude that the choice of imputation method can significantly influence accuracy and fairness. This corroborates previous findings in e.g. [23, 26, 36] that the best imputation method for a given prediction task and setup is not always a clear choice.

# 7 Conclusion and Limitations

In this work, we investigated the impact of missing values on algorithmic fairness. We introduced adaptive algorithms that can be used with any preexisting fairness-intervention algorithm to achieve higher accuracy and fairness than impute-then-classify. The flexibility and consistency of our algorithms highlight the importance of preserving the information in the missing patterns in fair learning pipelines. We note, however, that using missingness information in a fair classifier is not immune to potential negative impacts. For example, an individual may be incentivized to purposefully hide data if their true values are less favorable than even NA. In missing pattern clustering, an individual may be classified less favorably or accurately than if a classifier from a different cluster were used. These scenarios highlight considerations with respect to individual and preference-based fairness notions [55] and the importance of making careful, informed choices for both the adaptive algorithm and the fairness intervention. Another potential challenge is if sensitive groups are not known prior to deployment and must be learned online. Nevertheless, we hope that our insights can inform future work in fair classification with missing values.

Data imputation remains commonly used in data science pipelines. However, the predictions output by ML models are dependent on the imputation method used. When considering single data points, this presents an important challenge from an individual fairness perspective, because the choice of imputation may arbitrarily affect an individual's outcome [13]. The effect of imputation choice on individual outcomes parallels the "crisis in justifiability" that arises from model multiplicity, where a model may be unjustifiable for an individual if another equally accurate model provides the individual a better outcome [5, 41]. A future work direction is examining how to choose an imputation given these challenges and how the user may be involved in making this choice. Additionally, increased classification error or outcome volatility may harm individuals with many missing values. This raises the critical question of whether to withhold classification for individuals with too many missing values and how to provide resources to unclassified individuals. Ultimately, the ubiquity of missing data in the real world means that new developments in fairness with missing values have the potential to be broadly impactful and create better, fairer societal outcomes.

## Acknowledgement

The authors would like to thank Prof. Haewon Jeong for her valuable input in the early stage of this project. This material is based upon work supported by the National Science Foundation under grants CAREER 1845852, CIF 2312667, FAI 2040880, CIF 1900750.

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

# A   Omitted Proofs

## A.1   Proof of Theorem 1

*Proof.* We prove this theorem via construction. We consider $\mathcal{X} = \{0, 1\}$ and construct the probability distribution $P_{S,X,Y}$ in the following way:

$$\Pr(Y = 0, X = 0|S = s) = \Pr(Y = 0, X = 1|S = s) = \frac{1 - \alpha_s}{2}, \quad \Pr(Y = 0, X = \text{NA}|S = s) = 0,$$

$$\Pr(Y = 1, X = 0|S = s) = \Pr(Y = 1, X = 1|S = s) = 0, \quad \Pr(Y = 1, X = \text{NA}|S = s) = \alpha_s,$$

and $\Pr(S = s) = q_s$. By the construction of $P_{S,X,Y}$, we have

$$\Pr(Y = 1, S = 0) = \alpha_0 q_0, \qquad\qquad \Pr(Y = 1, S = 1) = \alpha_1 q_1,$$
$$\Pr(Y = 0, S = 0) = (1 - \alpha_0)q_0, \qquad \Pr(Y = 0, S = 1) = (1 - \alpha_1)q_1.$$

For this probability distribution, we can compute

$$I(M; Y) = (1 - \alpha) \log \frac{1}{1 - \alpha} + \alpha \log \frac{1}{\alpha} = g(\alpha).$$

Consider a classifier $h$ such that $h(0) = h(1) = 0$ and $h(\text{NA}) = 1$. This classifier satisfies $\mathbb{E}\left[\mathbb{I}(h(X) = Y)\right] = 1$ and $\text{Disc}(h) = 0$ where $\text{Disc}(h)$ is the equalized odds constraint. As a result, $F_\epsilon(P_{S,X,Y}) = 1$ for any $\epsilon$. By symmetry, we assume $f_{\text{imp}}(\text{NA}) = 1$ without loss of generality. Therefore, we have

$$\Pr(Y = 0, \hat{X} = 0|S = s) = \frac{1 - \alpha_s}{2} \qquad \Pr(Y = 0, \hat{X} = 1|S = s) = \frac{1 - \alpha_s}{2},$$
$$\Pr(Y = 1, \hat{X} = 0|S = s) = 0, \qquad\qquad \Pr(Y = 1, \hat{X} = 1|S = s) = \alpha_s.$$

We represent a probabilistic classifier $h$ as

$$\Pr(\hat{Y} = 0|\hat{X} = 0) = p_0, \qquad \Pr(\hat{Y} = 1|\hat{X} = 0) = 1 - p_0,$$
$$\Pr(\hat{Y} = 0|\hat{X} = 1) = p_1, \qquad \Pr(\hat{Y} = 1|\hat{X} = 1) = 1 - p_1.$$

Consequently,

$$\Pr(\hat{Y} = 0|Y = 0, S = s) = \frac{p_0 + p_1}{2}, \qquad \Pr(\hat{Y} = 1|Y = 0, S = s) = 1 - \frac{p_0 + p_1}{2},$$
$$\Pr(\hat{Y} = 0|Y = 1, S = s) = p_1, \qquad\qquad \Pr(\hat{Y} = 1|Y = 1, S = s) = 1 - p_1,$$

which ensures that $\text{Disc}(h) = 0$ where $\text{Disc}(h)$ is the equalized odds constraint. Furthermore,

$$
\begin{aligned}
\mathbb{E}\left[\mathbb{I}(\hat{Y} = Y)\right] &= \Pr(\hat{Y} = Y) \\
&= \Pr(\hat{Y} = 1|Y = 1, S = 0)\Pr(Y = 1, S = 0) + \Pr(\hat{Y} = 1|Y = 1, S = 1)\Pr(Y = 1, S = 1) \\
&\quad + \Pr(\hat{Y} = 0|Y = 0, S = 0)\Pr(Y = 0, S = 0) + \Pr(\hat{Y} = 0|Y = 0, S = 1)\Pr(Y = 0, S = 1) \\
&= (1 - p_1)(\alpha_0 q_0 + \alpha_1 q_1) + \frac{p_0 + p_1}{2}(1 - \alpha_0 q_0 - \alpha_1 q_1).
\end{aligned}
$$

Note that $\alpha_0 q_0 + \alpha_1 q_1 = \alpha$. Hence,

$$
\begin{aligned}
\max_h \mathbb{E}\left[\mathbb{I}(\hat{Y} = Y)\right] &= \max_{p_0, p_1 \in [0,1]} (1 - p_1)\alpha + \frac{p_0 + p_1}{2}(1 - \alpha) \\
&= \max_{p_0, p_1 \in [0,1]} \alpha + \frac{1 - \alpha}{2}p_0 + \frac{1 - 3\alpha}{2}p_1.
\end{aligned}
$$

Since $\alpha < 1/3$, we have $1 - \alpha > 0$ and $1 - 3\alpha > 0$. As a result, the above objective function reaches its maximum $1 - \alpha$ when $p_0$ and $p_1$ are both equal to 1. In other words, $F_\epsilon(P_{S,\hat{X},Y}) = 1 - \alpha$. $\qquad\square$

## A.2 Proof of Lemma 1

*Proof.* Let $y \in \{0, 1\}$ and $s, s' \in \mathcal{S}$. Then

$$|\Pr(\mathcal{C}(X) = 1|Y = y, S = s) - \Pr(\mathcal{C}(X) = 1|Y = y, S = s')|$$

$$= \left|\sum_j p_j \Pr(\mathcal{C}_j(X) = 1|Y = y, S = s) - \sum_j p_j \Pr(\mathcal{C}_j(X) = 1|Y = y, S = s')\right|$$

$$\leq \sum_j p_j |\Pr(\mathcal{C}_j(X) = 1|Y = y, S = s) - \Pr(\mathcal{C}_j(X) = 1|Y = y, S = s')|$$

$$\leq \sum_j p_j \varepsilon = \varepsilon.$$

$\square$

## B Information-Theoretic Interpretation of Adding Missing Indicators

We provide an information-theoretic perspective for how adding misisng indicators preserves information in the missing patterns. Let $Y$ be the unobserved predictive label we wish to recover, $X$ a variable representing the input features, and $\hat{X}$ the input features under an imputation function $f_{\text{imp}}$. By the data processing inequality, $I(Y; X) \geq I(Y; \hat{X})$, showing that imputation can lose information about $Y$. In contrast, because we can recover $X$ perfectly from $(\hat{X}, M)$ without using any information from $Y$, $I(Y; X) = I(Y; (\hat{X}, M))$, i.e. all of the original information about $Y$ is preserved.

Furthermore, we can use Fano's inequality [Theorem 6.3 in 46] to relate the information loss from imputation to classification error probability. Let $h$ be a classifier and denote the classification error as $p_e(X) = P(Y \neq h(X))$. Fano's inequality states that

$$H_2(p_e(X)) + p_e(X) \log(|\mathcal{Y}| - 1) \geq H(Y|X).$$

where $\mathcal{Y}$ denotes the support of $Y$, $H_2(p) = -p \log_2 p - (1 - p) \log_2(1 - p)$ is the binary entopy function, and $H(Y|X) = -\sum_{i,j} p(y_i, x_j) \log p(y_i|x_j)$ is the conditional entropy. Since $Y \in \{0, 1\}$, the inequality reduces to

$$H_2(p_e(X)) \geq H(Y|X) \implies p_e(X) \geq H(Y|X) - (\log_2 3 - 1)$$

since $H_2(p) \leq p + \log_2 3 - 1$ with equality achieved at $p = \frac{1}{3}$. This is a lower bound on $p_e(X)$ in terms of $H(Y|X)$. Now, we can compare $H(Y|\hat{X})$ and $H(Y|(\hat{X}, M))$. By above, we have

$$I(Y; (\hat{X}, M)) \geq I(Y; \hat{X})$$
$$H(Y) - H(Y|(\hat{X}, M)) \geq H(Y) - H(Y|\hat{X})$$
$$H(Y|(\hat{X}, M)) \leq H(Y|\hat{X}).$$

Therefore, Fano's inequality gives a better lower bound for $p_e(\hat{X}, M)$ than for $p_e(\hat{X})$. We conclude that a classifier trained from adding missing indicators is more likely to achieve higher accuracy under the same group fairness constraint than a classifier trained using impute-then-classify.

## C More Details on Algorithms

We include a figure illustrating the three adaptive algorithms presented in Section 4 for fair linear classifiers and the `FairMissBag` algorithm presented in Section 5. We also provide pseudocode for the recursive partitioning procedure in the missing pattern clustering algorithm and the fair uniform resampling procedure in `FairMissBag`.

## C.1 Illustration of adaptive algorithms

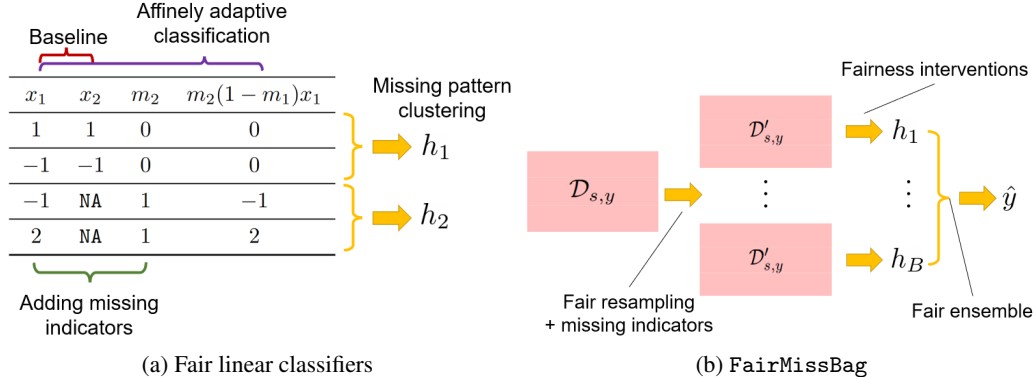

(a) Fair linear classifiers

(b) `FairMissBag`

Figure 4: Illustration of the three adaptive algorithms for fair linear classifiers (left) and `FairMissBag` (right). Adding missing indicators and affinely adaptive classification encode missing value patterns via additional input features, while missing pattern clustering partitions the dataset based on missing patterns. `FairMissBag` is a three-step algorithm consisting of fair resampling, using a fairness intervention to train a fair classifier on each resampled dataset, and obtaining a prediction from the ensemble of fair classifiers.

## C.2 Recursive partitioning algorithm for missing pattern clustering

We begin with a single cluster containing the entire dataset. At each step, we split a cluster on a feature $j^* \in G(\mathcal{M}_q)$ that minimizes the objective function $L(\mathcal{M}_q, j)$, where $G(\mathcal{M}_q)$ denotes the set of features that satisfy the bounded representation and minimum cluster size constraints.

---

**Algorithm 1** Recursive partitioning of missing patterns.

---

**Input:** $\mathcal{D} = \{x_i, s_i, y_i\}_{i=1}^n$
**Initialize:** partition $\mathcal{P} = \varnothing, \mathcal{P}' = \mathcal{M}$
**while** $\mathcal{P}' \neq \varnothing$ **do**
    $\mathcal{M}_q \leftarrow \mathcal{P}'[0]$
    $\mathcal{P}' \leftarrow \mathcal{P}' \setminus \mathcal{M}_q$
    **if** $G(\mathcal{M}_q) = \varnothing$ **then**
        $\mathcal{P} \leftarrow \mathcal{P} \cup \mathcal{M}_q$
        **continue**
    **end if**
    $j^* \leftarrow \arg\min_{G(\mathcal{M}_q)} L(\mathcal{M}_q, j)$
    **if** $L(\mathcal{M}_q, j^*) < \min_{h \in \mathcal{H}} \sum_{i \in \mathcal{I}_q} \ell(y_i, h(\boldsymbol{x}_i))$ **then**
        $\mathcal{P}' \leftarrow \mathcal{P}' \cup \{\mathcal{M}_q^{j0}, \mathcal{M}_q^{j1}\}$
    **else**
        $\mathcal{P} \leftarrow \mathcal{P} \cup \mathcal{M}_q$
    **end if**
**end while**
**Return:** $\mathcal{P} = \{\mathcal{M}_q\}_{q=1}^Q$

---

## C.3 Fair uniform resampling algorithm

The data is separated into groups based on sensitive attribute and label. From each group, we make a new group of the same size by sampling data points uniformly with replacement. These groups are combined to create a new subsample of the same size as the original dataset.

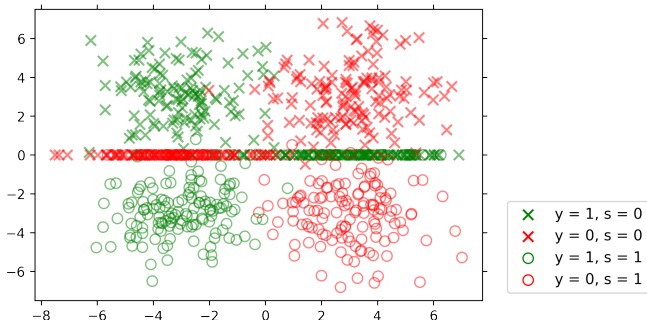

Figure 5: Plot of the synthetic data. The points on the $x$-axis represent data with $X_2$ missing.

---

**Algorithm 2** Single iteration of fair uniform resampling.

---

**Input:** $\mathcal{D} = \{\boldsymbol{x}_i, s_i, y_i\}_{i=1}^n$
**Initialize:** $\mathcal{D}' = \varnothing$
**for** $s \in \{0, 1\}$ **do**
    **for** $y \in \{0, 1\}$ **do**
        $\mathcal{D}_{s,y} \leftarrow \{\boldsymbol{x}_i, s_i, y_i \in \mathcal{D} | s_i = s, y_i = y\}$
        Sample $|\mathcal{D}_{s,y}|$ times from $\mathcal{D}_{s,y}$ uniformly with replacement and append to $\mathcal{D}'$
    **end for**
**end for**
Add missing indicators $\boldsymbol{m}$ to $\mathcal{D}'$, apply imputation method $f_{\text{imp}}$
**Return:** Resampled dataset $\mathcal{D}'$

---

# D  Details on the Experimental Results

We provide a description of the synthetic dataset mentioned at the end of Section 6.2 and a description of COMPAS/Adult. We also summarize hyperparameters and implementation details.

## D.1  Description of synthetic data

We create a synthetic dataset by sampling data points from a known distribution and generating artificial missing values. The dataset contains two features, $X_1$ and $X_2$, with binary sensitive attribute $S$ and binary label $Y$. There are 2400 data points in total. $X_1$ is present for all data points, while $X_2$ is missing in 800 of the data points. For each combination of predictive label and sensitive attribute, we sample data points from a different, fixed distribution (bivariate Normal for data points with $X_2$ present and Gaussian for data points with $X_2$ missing). The distribution of the data points is summarized in table 1, and we visualize 800 of the data points in figure 5.

| $X_2$ present | | |
|---|---|---|
| Group | Distribution | $N$ |
| $P(X_1, X_2 \| Y = 1, S = 1)$ | $\mathcal{N}\left(\begin{bmatrix} -3 \\ -3 \end{bmatrix}, \begin{bmatrix} 2 & 0 \\ 0 & 2 \end{bmatrix}\right)$ | 400 |
| $P(X_1, X_2 \| Y = 1, S = 0)$ | $\mathcal{N}\left(\begin{bmatrix} -3 \\ +3 \end{bmatrix}, \begin{bmatrix} 2 & 0 \\ 0 & 2 \end{bmatrix}\right)$ | 400 |
| $P(X_1, X_2 \| Y = 0, S = 1)$ | $\mathcal{N}\left(\begin{bmatrix} +3 \\ -3 \end{bmatrix}, \begin{bmatrix} 2 & 0 \\ 0 & 2 \end{bmatrix}\right)$ | 400 |
| $P(X_1, X_2 \| Y = 0, S = 0)$ | $\mathcal{N}\left(\begin{bmatrix} +3 \\ +3 \end{bmatrix}, \begin{bmatrix} 2 & 0 \\ 0 & 2 \end{bmatrix}\right)$ | 400 |

| $X_2$ missing | | |
|---|---|---|
| Group | Distribution | $N$ |
| $P(X_1 \| Y = 1, S = 1)$ | $\mathcal{N}(3, 3)$ | 100 |
| $P(X_1 \| Y = 1, S = 0)$ | $\mathcal{N}(3, 3)$ | 300 |
| $P(X_1 \| Y = 0, S = 1)$ | $\mathcal{N}(-3, 3)$ | 100 |
| $P(X_1 \| Y = 0, S = 0)$ | $\mathcal{N}(-3, 3)$ | 300 |

Table 1: Distribution of the synthetic data. $N$ denotes the number of data points sampled from the group distribution.

## D.2 Description of COMPAS and Adult datasets

**COMPAS**. The COMPAS (Correctional Offender Management Profiling for Alternative Sanctions) software is a proprietary tool that is used in the criminal justice system in the United States to predict recidivism. COMPAS uses a defendant's criminal history along with other features such as demographic information to assign a score indicating the predicted risk of the defendant recidivating within a two-year period. Racial discrimination in the COMPAS algorithm's predictions has been highly publicized and analyzed [11, 16, 34].

We restrict our experiments with the COMPAS dataset to the set of White and Black defendants. The predictive features we use are age (three categories of less than 25, 25 to 45, and over 45), race, sex, normalized number of prior convictions, and charge degree (misdemeanor/felony). The predictive label is a binary indicator for whether the defendant recidivated in a two-year period, and the sensitive attribute is race ($S = 0$ for Black defendants, $S = 1$ for White defendants). We additionally balance the dataset so that the number of Black and White defendants in the data are equal, yielding 4206 data points in the dataset.

**Adult**. The Adult dataset is a subset of the data from the 1994 Census. The predictive label is a indicator for whether a person's annual income is over $50,000, and the sensitive attribute is gender. We use all of the 14 features in the original dataset except fnlwgt (see [12]). We first balance the dataset by the predictive label and then by gender, yielding 6530 points in the dataset. All features are scaled to be between 0 and 1. We additionally use IPUMS Adult, a reconstructed superset of the original Adult dataset [9, 15], which contains 11996 points after balancing for predictive label and gender.

**Missing value generation**. Since COMPAS is a complete dataset and Adult has a very low percentage of missing values, we generate missing values synthetically. For each dataset, we create three datasets with missing values corresponding to MCAR, MAR, and MNAR to capture the range of possible missing mechanisms. For each missing mechanism, we choose one or more features to contain missing values. Then, we randomly replace some of the values of the feature(s) with `NA` with a given probability. Other than the MCAR setting, we allow the probability to vary depending on the value of one of more features in the data or a binary indicator corresponding to the feature, including the missing feature(s). Notably, for MNAR, we allow the missingness of a feature to depend on the predictive label to simulate the scenario where the missingness pattern and label are linked. Statistics for missing values in COMPAS and Adult are provided in Table 2.

|  | COMPAS | | | |
| --- | --- | --- | --- | --- |
| Mechanism | Missing feature | $I$ | $\Pr(\text{MS}|I = 0)$ | $\Pr(\text{MS}|I = 1)$ |
| MCAR | priors_count | N/A | 0.2 | 0.2 |
| MAR | priors_count | sex | 0.1 | 0.4 |
| MNAR | priors_count | two_year_recid | 0.1 | 0.4 |
|  | sex | sex | 0.1 | 0.4 |

|  | Adult | | | |
| --- | --- | --- | --- | --- |
| Mechanism | Missing feature | $I$ | $\Pr(\text{MS}|I = 0)$ | $\Pr(\text{MS}|I = 1)$ |
| MCAR | age | N/A | 0.2 | 0.2 |
| MAR | occupation | gender | 0.4 | 0.1 |
| MNAR | capital-gain | income | 0.3 | 0.1 |
|  | workclass | $\mathbb{I}[\text{age} < 0.2]$ | 0.1 | 0.4 |

Table 2: Missing value statistics for COMPAS and Adult. $I$ denotes the feature or indicator on which the probability of missingness depends. As an example, for COMPAS in the MAR setting, the missing feature is priors_count; the probability of priors_count being missing is $0.1$ among women (sex = 0) and $0.4$ among men (sex = 1).

### D.3 Hyperparameters

For algorithms with tunable hyperparameters used in the experiments, we report the values of the hyperparameters that were tested in Table 3.

| Algorithm | Hyperparameter | Synthetic | COMPAS/Adult | HSLS |
|---|---|---|---|---|
| `DispMistreatment` | $\tau$ | $\{0.01, 0.1, 1, 10, 100\}$ | $\{0.01, 0.1, 1, 2, 5, 10\}$ | $\{0.01, 0.1, 1, 10, 100\}$ |
| `FairProjection` | tol | $\{0, 0.001, 0.01, 0.1, 0.5\}$ | $\{0, 0.001, 0.01, 0.05, 0.1, 0.5, 1\}$ | $\{0, 0.001, 0.01, 0.1, 0.5\}$ |
| `Reduction` | $\varepsilon$ | N/A | N/A | $\{0.001, 0.005, 0.01, 0.02, 0.05, 0.1, 0.2, 0.5, 1, 2\}$ |
| `ROC` | $\varepsilon$ | N/A | N/A | $\{0.001, 0.003, 0.01, 0.03, 0.1, 0.3, 1\}$ |
| Missing pattern clustering | $k$ | 1 | N/A | $\{500, 1000, 1500, 2000, 4000\}$ |
| | $(\alpha, \beta)$ | $(1, 0)$ | N/A | $(0.6, 0.3)$ |

Table 3: Summary of hyperparameters tested for the experiments with linear fair classifiers. The `DisparateMistreatment`, `FairProjection`, `Reduction`, and `ROC` fairness-intervention algorithms have tunable hyperparameters, as well as the missing pattern clustering adaptive algorithm.

| Algorithm | Hyperparameter | Values |
|---|---|---|
| `FairProjection` | tol | $\{0.000, 0.001, 0.01, 0.1\}$ |
| `Reduction` | $\varepsilon$ | $\{0.001, 0.01, 0.1, 0.5, 1\}$ |
| `FairMissBag` | # bags | 20 (`FairProjection`) |
| | | 10 (`Reduction`, `EqOdds`) |

Table 4: Summary of hyperparameters tested for the experiments with general fair classifiers. The `FairProjection` and `Reduction` fairness-intervention algorithms have tunable hyperparameters, as does `FairMissBag`.

### D.4 Implementation details

We run `DispMistreatment` with a FNR difference constraint, `FairProjection`, `Reduction`, `EqOdds`, `ROC` with an equalized odds/mean equalized odds (EO/MEO) constraint, and `Leveraging` with an equality of opportunity constraint.

We implement `DispMistreatment` using the code at https://github.com/mbilalzafar/fair-classification.For `FairProjection`, we use code from https://github.com/HsiangHsu/Fair-Projection. The original code for `Leveraging` is found at https://github.com/lucaoneto/NIPS2019_Fairness; we use a Python version of the code included in the `FairProjection` repository. `Reduction`, `EqOdds`, and `ROC` are implemented using the AIF360 library [3]. All experiments were run on a personal computer with 4 CPU cores and 16GB memory.

## E Additional Experimental Results

### E.1 Linear fair classifiers

**Synthetic data.** Figure 6 displays fairness-accuracy curves for the synthetic dataset. We see that for both `DispMistreatment` and `FairProjection`, missing pattern clustering (Section 4.3) achieves near-perfect accuracy with very little discrimination. In contrast, the baseline performs much worse in both cases; for `DispMistreatment`, the baseline classifier only achieves the same fairness as missing pattern clustering with an accuracy that is essentially no better than a random guess. Adding missing indicators outperforms the baseline for `DispMistreatment` but remains significantly worse than missing pattern clustering, and performs nearly identically to the baseline in `FairProjection`.

These results make intuitive sense given the distribution of the synthetic data. $\mathbb{I}[X_1 < 0]$ is a near-perfect linear classifier for the data points with $X_2$ present, and $\mathbb{I}[X_1 > 0]$ is a near-perfect linear classifier for the data points with $X_2$ missing. However, if we use impute-then-classify with zero imputation, a baseline linear classifier will perform poorly on the data points with $X_2$ missing. Moreover, because the $S = 0$ sensitive group has many more such points, the group will likely suffer

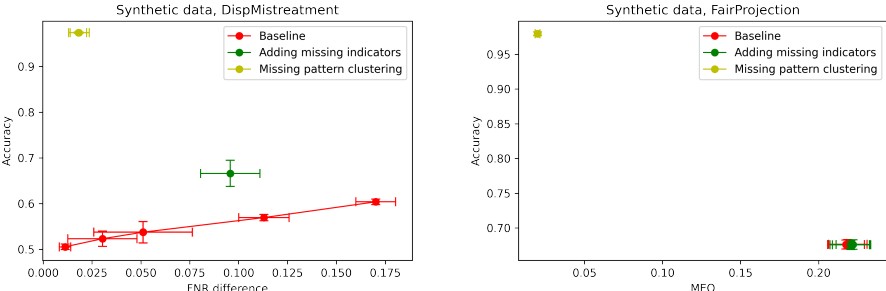

Figure 6: Comparison of adaptive fairness-intervention algorithms on the synthetic dataset, using `DispMistreatment` (left) and `FairProjection` (right). Error bars depict the standard error of 5 runs with different train-test splits.

from higher FNR and FPR, increasing the discrimination. Furthermore, this is true even if a missing indicator for $X_2$ is added, as the indicator can only provide an adjustment for the bias and not the coefficient of $X_1$. However, if missing pattern clustering is used, separate linear classifiers can be trained on each cluster to reach very high accuracy and fairness.

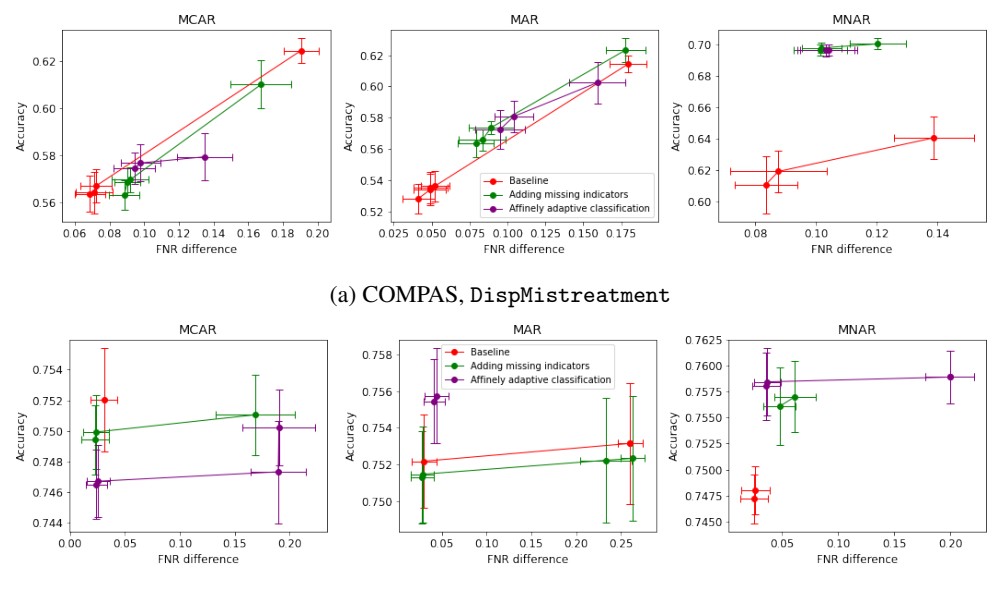

(a) COMPAS, `DispMistreatment`

(b) Adult, `DispMistreatment`

Figure 7: Comparison of adaptive fairness-intervention algorithms on COMPAS and Adult using `DispMistreatment`. For each of COMPAS and Adult, we generate three datasets with missing values corresponding to MCAR, MAR, and MNAR mechanisms (Appendix D.2). Error bars depict the standard error of 10 runs with different train-test splits.

**HSLS.** Figure 8 displays the fairness-accuracy curves for `Reduction`, `EqOdds`, `ROC`, `Leveraging`. We see that adding missing indicators improves the fairness-accuracy tradeoff for all fairness interventions except `EqOdds`.

We report additional results for the missing pattern clustering algorithm on the HSLS dataset. As indicated above, we tested minimum cluster sizes of 500, 1000, 1500, 2000, and 4000. The average number of clusters produced from missing pattern clustering across all trials was 2.5 for $k = 500$, 1.8 for $k = 1000$ and $k = 1500$, 1.5 for $k = 2000$, and 1.3 for $k = 4000$. The highest number of clusters obtained from a single trial was 5. As the minimum cluster size is increased, the average number of clusters decreases monotonically as expected. The low average cluster counts indicate that the missing pattern clustering algorithm was not always able to find a viable split of the missing patterns

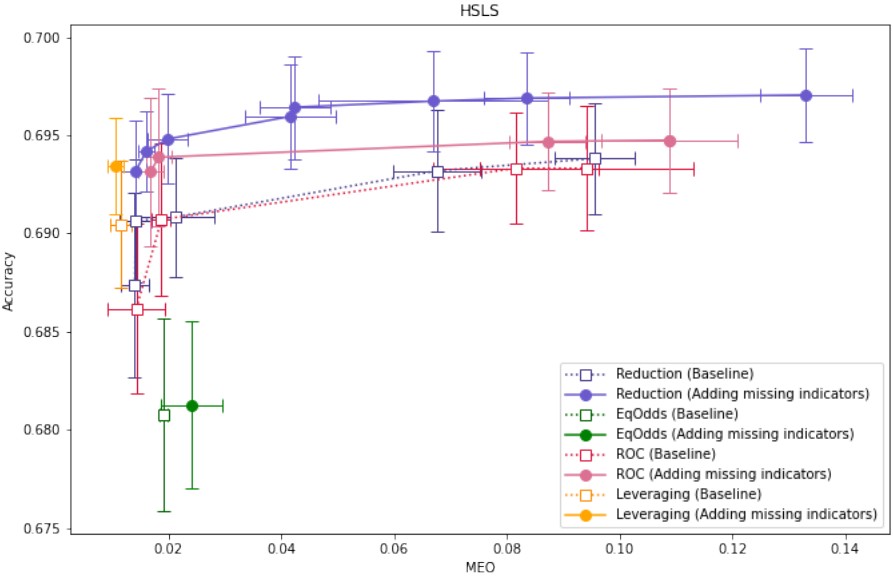

Figure 8: Comparison of adding missing indicators (Section 4.1) with baseline impute-then-classify on HSLS, using `Reduction`, `EqOdds`, `ROC`, and `Leveraging`. Error bars depict the standard error of 15 runs with different train-test splits. .

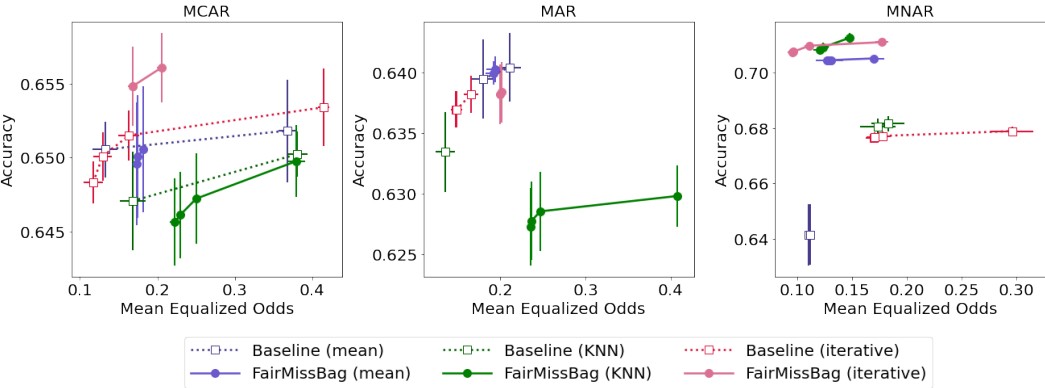

Figure 9: Results for `FairMissBag` on COMPAS using `FairProjection`. Using COMPAS, we generate three datasets with missing values corresponding to MCAR, MAR and MNAR mechanisms (Section D.2). Fairness-accuracy curves for mean, KNN, and iterative imputation are shown. Error bars depict the standard error of 5 runs with different train-test splits.

(see the discussion of the semi-synthetic experimental results in Section 6.2 for an explanation on this behavior).

## E.2 Non-linear fair classifiers

**COMPAS**. Figure 9 displays the results of the `FairMissBag` experiment on COMPAS. When missing values are MNAR, `FairMissBag` shows large improvements in both accuracy and fairness for all three imputation mechanisms.

