# OpenReview forum: "Adapting Fairness Interventions to Missing Values"
_NeurIPS.cc/2023/Conference — NeurIPS 2023 poster_

### Official Review · Reviewer_SEin · 2023-07-05

**Soundness:** 3 good
**Presentation:** 2 fair
**Contribution:** 2 fair
**Rating:** 5
**Confidence:** 4

**Summary:**

This paper presents an information-theoretic finding that reveals the fundamental limitation of impute-then-classify approaches when considering fairness-accuracy tradeoffs. Additionally, it introduces three techniques for addressing missing features within the framework of linear fair classifiers, as well as an ensemble method for non-linear counterparts. One notable aspect of these developments is their ability to capture missing pattern information, which is overlooked by impute-then-classify algorithms. Furthermore, the paper presents experimental results that demonstrate the superior tradeoff performances of the proposed methods, particularly when dealing with datasets that exhibit prominent missing patterns.

**Strengths:**

S1. The paper focuses on a significant issue that arises in numerous applications.

S2. By utilizing the concept of mutual information, the paper reveals the fundamental limitation of impute-then-classify methods.

S3. The proposed methods effectively harness the information embedded within missing patterns.

**Weaknesses:**

W1. I believe that scenarios where sensitive attributes are missing present more practical relevance, importance, and challenges compared to scenarios where features are missing. Although the authors mention the possibility of extending their findings to such settings in the conclusion section, the specific details of this extension remain unclear as the computation of fairness constraints relies on knowledge of sensitive attributes.

W2. The main inspiration for this paper appears to be derived from [3]. While the main contribution of this paper lies in its adaptation to the fairness context, it does not take into account the scenario where sensitive attributes are missing. Exploring this more challenging setting may open up the opportunity for a distinct idea to be explored.

W3. The paper introduces several methods for linear and non-linear settings, suggesting that the choice should depend on the data distribution. However, it does not provide concrete guidelines as to how to make the choice.

W4. Theorem 1 looks interesting, but the main body of the paper lacks technical discussion, not even including proof sketch.

**Questions:**

In line of W1, can you provide in-depth discussion on the extension of the sensitive-attribute-missing scenario?

In light of W4: any intuition why the fundamental degradation of accuracy is expressed in terms of mutual information?

**Limitations:**

Please see Weaknesses in the above.

---

> ### Author Rebuttal · Authors · 2023-08-09
>
> We thank the reviewer for their careful read of our paper and constructive comments!
>
> ---
> **W1. I believe that scenarios where sensitive attributes are missing present more practical relevance, importance, and challenges compared to scenarios where features are missing. Although the authors mention the possibility of extending their findings to such settings in the conclusion section, the specific details of this extension remain unclear as the computation of fairness constraints relies on knowledge of sensitive attributes.**
>
> A1. We completely agree that the problem of missing group (sensitive) attributes is of significant practical importance. Indeed, there is a growing body of research (Kallus et al. 2022, Zhang and Long 2021) in fair ML dedicated to addressing this issue. However, we also emphasize that the problem of missing input features is equally crucial and widespread, yet less studied. For instance, in the HSLS dataset used in our experiment, 35.5% of White and Asian students did not report their secondary caregiver’s highest level of education; this proportion increases to 51.0% for underrepresented minority students. Missingness patterns can vary significantly across population groups and hinder fairness-accuracy performance if not adequately accounted for. Unfortunately, the matter of missing input features has not received adequate attention in fair ML literature, with the majority of interventions assuming complete input features.
>
> The primary objective of this paper is to bridge this gap and offer a comprehensive study encompassing theory and algorithms for training fair classifiers in the presence of missing input feature values, particularly when missingness patterns vary per population group (as observed in practice). We aim to shed light on this overlooked aspect and propose solutions to challenges arising from disparate missingness patterns. To address your concerns, we will revise our abstract and introduction to provide a clearer presentation of our setup and discuss the issue of missing group attributes by doing a more thorough job acknowledging this important line of work.
>
> References:
>
> Kallus, N., Mao, X., & Zhou, A. (2022). Assessing algorithmic fairness with unobserved protected class using data combination.
>
> Zhang, Y., & Long, Q. (2021). Assessing fairness in the presence of missing data.
>
> ---
> **W2. The main inspiration for this paper appears to be derived from [3]. While the main contribution of this paper lies in its adaptation to the fairness context, it does not take into account the scenario where sensitive attributes are missing.**
>
> A2. Please refer to our response to W1. We would like to emphasize the differences between our study and [3]. Not only do we adapt their algorithms for the fairness context, but we also develop a *new* universal adaptive algorithm outlined in Section 5 which allows for the adaptation of *any* group-fairness intervention to account for missing values. In contrast, [3] focuses on linear models only. In Section 3, we also derive a theorem that articulates the fairness risks associated with training on imputed data. Furthermore, our work involves extensive numerical experiments using state-of-the-art fairness interventions across multiple benchmark datasets (see Appendix E).
>
> ---
> **W3. The paper introduces several methods for linear and non-linear settings, suggesting that the choice should depend on the data distribution. However, it does not provide concrete guidelines as to how to make the choice.**
>
> A3. We thank the reviewer for raising this point! Please refer to our response to Reviewer oma2’s Q1 and Q2.
>
> ---
> **W4. Theorem 1 looks interesting, but the main body of the paper lacks technical discussion, not even including proof sketch.**
>
> A4. We are glad to learn that you find Theorem 1 interesting! This theorem was established by creating a specific data generating distribution and identifying the optimal classifiers for both data with missing values and imputed data. We intend to add a sketch of the proof in order to provide further clarity and enhance understanding of the theorem's underlying rationale.
>
> ---
> **Q1. In line of W1, can you provide in-depth discussion on the extension of the sensitive-attribute-missing scenario?**
>
> A5. Absolutely – please refer to section (a) of our answer to Q1 in the global response.
>
> ---
> **Q2. In light of W4: any intuition why the fundamental degradation of accuracy is expressed in terms of mutual information?**
>
> A6. The underlying rationale for Theorem 1 is that as missing patterns provide more information about the predictive label, the fairness-accuracy performance of impute-then-classify deteriorates. This occurs as imputing the data loses information about the label that was contributed by the missing patterns. We use mutual information to measure the extent of this information, since mutual information is a standard metric for assessing the dependence between two random variables. It also enables us to apply Fano’s inequality to relate the information loss from imputation to classification error probability (see Appendix B).

---

### Official Review · Reviewer_RyBS · 2023-07-06

**Soundness:** 4 excellent
**Presentation:** 3 good
**Contribution:** 4 excellent
**Rating:** 6
**Confidence:** 4

**Summary:**

This paper investigates the impact of missing values on algorithmic fairness and highlights the limitations of the commonly used "impute-then-classify" approach. The authors propose algorithms that preserve the information encoded within missing patterns, leading to improved fairness and accuracy.

**Strengths:**

1. The paper theoretically shows that for the fairness measure of equalized odds, impute-then-classify can significantly reduce the performance. Furthermore, it is also shown that the reduction in the performance grows with the mutual information between the missing pattern and the labels.

2. The paper proposes 3 methods to handle missing values for linear fair classifiers by encoding the missing value patterns. These methods are interpretable and can be combined with any preexisting fairness intervention method including in-processing and post-processing methods.

3. The paper extensively evaluates the proposed method on synthetic data as well as real data. The authors also show the superiorty of the proposed methods for linear setting for the MNAR missing pattern.

Overall, the paper is mostly clear and has original ideas.

**Weaknesses:**

1. The proposed method is only applicable to fair classification and when the group attributes are discrete. Furthermore, the approach allows missingness only in the non-group attribute input features, i.e., the method requires the group attribute and the labels to be fully observed. It might be useful to extend the method for fair regression and for missingness in group attribute and labels.

2. The results in the paper focus only on a single measure of fairness, i.e., equalized odds. (By the way, the MEO abbreviation in the figures in the main body should be expanded in the caption when first used.) It might be useful to extend the method for other notions of fairness and provide analogous empirical evaluation.

**Questions:**

1. What is the definition of the function h in equation (1)? Also, Disc(h) should be Disc(h(X)) in equation (1).

2. Can the authors elaborate on how their framework can be extended to multiaccuracy and multicalibration notions of fairness? Are there any empirical evaluations for these settings that the authors have performed?

3. It would help to show empirically how the proposed method performs with respect to the approach of Jeong et al and also state the corresponding runtimes. Also, what does MIA in line 96 stand for?

4. How are the B datasets created in lines 249-250? Line 248 indicates there is only one combined dataset.

5. The algorithm in Section 5 seems closely connected to the idea of using bootstraped subsamples proposed in 'Group Fairness with Uncertainty in Sensitive Attributes'. Could the authors clarify on the similarities and the differences? It might be useful to talk about this in related work.

6. For the HSLS dataset, why do the authors only consider datapoints where race and 9th grade math test score are present?

7. The focus of the paper is on algorithm design for the case of missing non-group attribute input features. I would advise the authors to state this right in the abstract and also talk about it in the introduction. A lot of literature on algorithmic fairness focuses on missing group attribute input features and it is easy to inherently assume that this paper does the same until one reaches line 107.

**Limitations:**

The authors talk about the limitations of their work in Section 7. While they briefly talk about potential negative societal impact of data imputation in Section 7 too, I encourage the authors to talk about the potential negative impact of their methodology too.

---

> ### Author Rebuttal · Authors · 2023-08-09
>
> We thank the reviewer for the thoughtful review and for appreciating the merits of the work!
>
> **W1. The proposed method is only applicable to fair classification and when the group attributes are discrete. Furthermore, the approach allows missingness only in the non-group attribute input features, i.e., the method requires the group attribute and the labels to be fully observed. It might be useful to extend the method for fair regression and for missingness in group attribute and labels.**
>
> A1. This is a great point! Please refer to section (a) in our answer to Q1 in the global response.
>
>
> ---
> **W2. The results in the paper focus only on a single measure of fairness, i.e., equalized odds. (By the way, the MEO abbreviation in the figures in the main body should be expanded in the caption when first used.) It might be useful to extend the method for other notions of fairness and provide analogous empirical evaluation.**
>
> A2. Thank you for the point of clarification. We note that while the empirical evaluations with FairProjection use equalized odds, we use equality of opportunity (FNR difference) for the results with DispMistreatment. These are primary fairness measures supported by the FairProjection and DispMistreatment fairness interventions. Additionally, while the fairness guarantee for the ensemble algorithm in Section 5 is presented for equalized odds, the guarantee also holds for other fairness measures such as statistical parity and equality of opportunity. We will update the caption with the full name of MEO in the revised paper.
>
> ---
> **Q1. Clarification of equation (1).**
>
> A3. The function $h: \mathcal{X}\to \mathcal{Y}$ represents a classifier, predicting the label $y$ based on input features $x$. We will make this clear in the revised paper and replace $Disc(h)$ with $Disc(h(X))$ as you suggested.
>
> ---
> **Q2. Extending framework to multiaccuracy and multicalibration notions of fairness.**
>
> A4. Thanks for pointing this out –  please refer to section (b) in our answer to Q1 in the global response where we address this point.
>
> ---
> **Q3. Comparing to Jeong et al; definition of MIA**
>
> A5. This is a great point. We encountered some challenges when attempting to compare our proposed methods to the FairMIPForest algorithm in Jeong et al., in that 1) the code for FairMIPForest does not support MEO and 2) DispMistreatment is a linear fairness intervention, hindering a sound comparison between FairMIPForest and either DispMistreatment or FairProjection. MIA stands for the “missingness incorporated in attribute” approach for training decision trees with missing values (see Twala et al. 2008). We will provide the full name in the revised paper.
>
> References:
>
> Twala, B. E., Jones, M. C., & Hand, D. J. (2008). Good methods for coping with missing data in decision trees.
>
> ---
> **Q4. How are the B datasets created in lines 249-250? Line 248 indicates there is only one combined dataset.**
>
> A6. Lines 247-249 describe the procedure to draw a single resampled dataset from the original dataset. The B datasets are created by repeating this procedure B times. We will clarify this in the revised paper.
>
> ---
> **Q5. Comparison to Bootstrap-S algorithm in Shah et al.**
>
> A7. Thank you for bringing this work to our attention. Our algorithm in Section 5 and the Bootstrap-S algorithm presented in Shah et al. are indeed similar in using bootstrapping to satisfy strengthened fairness conditions. The key difference is that our algorithm explicitly uses (known) sensitive attributes and labels when drawing subsamples to ensure sample diversity. We will include this comparison in the related work section in the updated paper.
>
> ---
> **Q6. For the HSLS dataset, why do the authors only consider datapoints where race and 9th grade math test score are present?**
>
> A8. This is a great question. For HSLS, we used race as the group attribute and 9th grade math test score as the label. As mentioned above, the fairness interventions with which our proposed methods are used may require knowledge of the group attribute, and our empirical results used interventions that use knowledge of the group attribute and label to calculate fairness metrics.
>
> ---
> **Q7. Clarifying the focus of the paper in the abstract and introduction.**
>
> A9. Thank you for highlighting this issue. We will ensure that both the abstract and introduction clearly state the context being considered in this paper.
>
> ---
> **Q8. Elaborating on potential negative impact of methodology in Section 7.**
>
> A10. We thank the reviewer for raising this important point. We acknowledge that using missingness information in a fair classifier may have potential negative impacts. For example, an individual may be incentivized to purposefully hide data if their true values are less favorable than even \texttt{NA}. In missing pattern clustering, an individual may be classified less favorably or accurately than a classifier from a different cluster. These scenarios highlight 1) important considerations with respect to individual and preference-based fairness notions (cf. Ustun et al. 2019), and 2) the importance of carefully weighing the advantages and disadvantages of each of our proposed methods prior to use. We will elaborate on these issues in the revised paper.
>
> Reference:
>
> Ustun, B., Liu, Y., & Parkes, D. (2019). Fairness without harm: Decoupled classifiers with preference guarantees.

---

> > ### Comment · Reviewer_RyBS · 2023-08-15
> >
> > Thank you for your detailed response. I've also gone through the reviews from other reviewers. Here are a few comments regarding your response:
> >
> > W1. I recognize the distinction between challenges arising from missing input features versus missing sensitive attributes. It might be beneficial for the authors to acknowledge this limitation explicitly, emphasizing that the framework considers missingness in features but not in labels or sensitive attributes. Also, there was no response addressing the fact that the method is limited to classification tasks (not regression) and discrete sensitive attributes (not continuous ones).
> >
> > W2. It would be useful if the use of FNR difference is highlighted somewhere in the paper. Clearly articulating the proposed method's applicability would be valuable.
> >
> > Q3. Could you provide further insight into why the linearity of the fairness intervention "DispMistreatment" hampers the comparison? Additionally, I'm curious about the compatibility of the FairMIPForest framework with other fairness interventions like Reduction, EqOdds, ROC, or Leveraging. Does the FairMIPForest framework support the assessment of FNR difference?
> >
> > Q6. Please mention the choice of variables as sensitive attributes and labels in the appropriate sections.

---

> > > ### Author Response · Authors · 2023-08-16
> > > **Thank you for your reply!**
> > >
> > > Thank you so much for your response and follow-up questions!
> > >
> > > ---
> > > **W1. I recognize the distinction between challenges arising from missing input features versus missing sensitive attributes. It might be beneficial for the authors to acknowledge this limitation explicitly, emphasizing that the framework considers missingness in features but not in labels or sensitive attributes. Also, there was no response addressing the fact that the method is limited to classification tasks (not regression) and discrete sensitive attributes (not continuous ones).**
> > >
> > > A1. Yes – we will clarify the scope of our framework in the updated paper and clearly state the limitation, as well as point to the references on handling missing sensitive attributes.
> > >
> > > Regarding fair regression, since the methods in Section 4 involve the input features only, we can apply them to fair regressors in an identical fashion as described for fair classifiers. Similarly, for continuous sensitive attributes, the methods can be applied in the same way provided the underlying fairness intervention is designed to handle continuous sensitive attributes. In general, the methods in section 4 depend on sensitive attributes and/or the target variable only to the extent of the adapted fairness intervention.
> > >
> > > Adapting the method in Section 5 to non-discrete labels (as is the case for fair regression) and/or continuous sensitive attributes is more complex because the resampling process uses the discrete nature of the sensitive attribute and label to preserve the joint distribution of the sensitive attribute and label in the subsampled dataset. We touched on this limitation in lines 355-358, but will make it explicit in a revised manuscript.
> > >
> > > ---
> > > **W2. It would be useful if the use of FNR difference is highlighted somewhere in the paper. Clearly articulating the proposed method's applicability would be valuable.**
> > >
> > > A2. Thank you for raising this point – we will mention the use of FNR difference prior to the experimental results involving the metric and clarify that our proposed methods work across several group fairness metrics including (but not limited to) equalized odds, equality of opportunity (FNR difference) and statistical parity.
> > >
> > > ---
> > > **Q3. Could you provide further insight into why the linearity of the fairness intervention "DispMistreatment" hampers the comparison? Additionally, I'm curious about the compatibility of the FairMIPForest framework with other fairness interventions like Reduction, EqOdds, ROC, or Leveraging. Does the FairMIPForest framework support the assessment of FNR difference?**
> > >
> > > A3. Absolutely. While FairMIPForest does support FNR difference, comparing a decision tree classifier such as FairMIPForest with a linear classifier such as DispMistreatment can be challenging because the models differ in their expressivity. For example, while FairMIPForest can capture nonlinearities in the data that cannot be captured by DispMistreatment, FairMIPForest is constrained by the depth of the decision tree. Additionally, we found that running FairMIPForest yielded a worse fairness-accuracy curve than DispMistreatment despite having a greater runtime. We believe the reason for this poor performance is because the available FairMIPForest code uses early stopping in the training process to account for the computational cost of solving the mixed-integer optimization in the algorithm’s implementation. Regarding the other fairness interventions benchmarked in Appendix E.1, only Leveraging supports assessing FNR difference and could thus be compared against FairMIPForest.
> > >
> > > ---
> > > **Q6. Please mention the choice of variables as sensitive attributes and labels in the appropriate sections.**
> > >
> > > A6. Will do. In section 6.1, we mention that for HSLS, the sensitive attribute is race and the label is a student’s test performance – we will clarify that the label refers to a student’s 9th grade test score.

---

> > > > ### Comment · Reviewer_RyBS · 2023-08-20
> > > >
> > > > Thank you for the clarifications!
> > > >
> > > > W1, W2, Q6 -- makes sense
> > > >
> > > > Q3 -- It would have been nice if the paper had included some comparisons with FairMIPForest using Leveraging but it may be infeasible at this stage.

---

> > > > > ### Author Response · Authors · 2023-08-20
> > > > >
> > > > > Thanks for your following up. We will make sure to include the promised changes in the revision (both in the main text and appendix). Additionally, we plan to conduct experiments comparing our adaptive algorithms that use Leveraging against FairMIPForest and will include the new experimental results in the revised paper. Thank you again for the insightful and constructive comments you provided!

---

### Official Review · Reviewer_ZFNz · 2023-07-07

**Soundness:** 3 good
**Presentation:** 3 good
**Contribution:** 3 good
**Rating:** 7
**Confidence:** 2

**Summary:**

This work investigates how different types of missing data affect algorithmic fairness, and provide algorithms that work to address this issue. Three types of missing data are considered: MCAR (missing data is independent of the observed and unobserved values), MAR (missing data depends on the observed values only, and MNAR (dependence of the missing data on the unobserved values). The contributions are (1) Theory showing that a model trained on imputed data (the classic impute-then-classify method) has unavoidable reduced performance; (2) Introduce strategies for adapting mitigation strategies in fair classification to missing data; and (3) provide an empirical analysis that supports the theory and compares against state-of-the-art fair classification algorithms that use impute-then-classify.

**Strengths:**

*ORIGINALITY.* I am not familiar with work in the missing data space, but looked through the related work section, skimmed a few of the works mentioned, and looked briefly at the literature on missing data in ML. This work seems to differ from previous contributions, and is adequately cited. The difference between the approach in this work and the approaches of previous work seems adequately explained.

*QUALITY.* The main contribution of this paper is to provide alternative methods to the classically used imputation-then-classify strategy for dealing with missing data in fair classification. This is a well-motivated problem, as imputation is used regularly when data is missing, and this work investigates the information lost when performing imputation, and provides an alternative, competitive strategy for dealing with missing data.

*CLARITY.* This work is clearly structured and well-written---I enjoyed the read!

*SIGNIFICANCE.* This work is a useful contribution to the literature on mitigating unfairness when values are missing from the feature vector (not including the sensitive attribute). The strategies for finding suitable models in this setting can be used for linear and non-linear classifiers, and the empirical analysis shows promising results that are competitive (and often outperform) methods that use imputation.

**Weaknesses:**

The fairness of models returned by the algorithms is not captured in the graphs in the main body of the work. The paper touts that training classifiers from imputed data can significantly worsen values of group fairness (and average accuracy), but their empirical analysis (in the main body) only compares the accuracy over datasets often used in fair classification.

*MINOR COMMENTS*
- The core contributions of this work seem to be applicable to general cases of missing data, not just fairness.
- Lines 114-117, providing a small example of the different types of reason for missing data could strengthen the description in this paragraph.
- Eqn 1 define the indicator function and distribution of interest in the expected value
- Fano's inequality (line 193) should be cited

**Questions:**

Could the authors elaborate on the group fairness achieved by their strategy vs the other baselines in the empirical analysis?

**Limitations:**

yes

---

> ### Author Rebuttal · Authors · 2023-08-09
>
> We appreciate the reviewer's thoughtful comments and we are glad to learn that you found our paper enjoyable to read!
>
> ---------
> **Q1. The fairness of models returned by the algorithms is not captured in the graphs in the main body of the work. The paper touts that training classifiers from imputed data can significantly worsen values of group fairness (and average accuracy), but their empirical analysis (in the main body) only compares the accuracy over datasets often used in fair classification.**
>
> A1. Thank you for raising this clarification point. We highlight that the x-axis of all the plots in the paper (Figs. 1-3 in the main text, and Figs. 6-9 in the Appendix) have a group fairness metric in the x-axis, and accuracy in the y-axis. Consequently, our empirical analysis explicitly compares accuracy and group fairness values achieved by different imputation strategies and fairness interventions across datasets and missing value patterns. Fairness-accuracy plots are standard visualizations for benchmarking fairness interventions (see, for example, [1] and [2]).
>
> ---------
> **Q2. The core contributions of this work seem to be applicable to general cases of missing data, not just fairness.**
>
> A2. Indeed, you are correct. A substantial number of our methods and algorithms are applicable to a range of supervised learning scenarios that involve missing values. We chose to concentrate on the fairness implications of missing data as it is a practical problem but has not received adequate attention in the fair ML literature.
>
> ---------
> **Q3. Lines 114-117, providing a small example of the different types of reason for missing data could strengthen the description in this paragraph.**
>
> A3. Yes, this is a great point! To illustrate the three missing mechanisms, we can use an example of student survey responses:
>
> MCAR: For each student, each question has an equal probability p of not being answered.
>
> MAR: Certain questions are more likely to be left blank depending on other factors. For example, lower-income students may be less likely to have taken expensive tests and consequently are more likely to leave questions about test scores blank.
>
> MNAR: The probability of a question being left blank depends on the true answer. For example, a student whose parents have not completed high school may leave a question on the parents’ highest degree blank.
>
> We will include this example in the revised paper.
>
> ---------
> **Q4. Eqn 1 define the indicator function and distribution of interest in the expected value.**
>
> A4. The indicator function takes the value of $1$ if $h(x) = y$; otherwise, it is 0. The distribution of interest is the data generating distribution $P_{X,Y,S}$ where $X$ is the input feature vector which may contain missing values; $Y$ is the label; and $S$ is the group attribute. We will include this clarification in the revised paper.
>
> ---------
> **Q5. Fano's inequality (line 193) should be cited**
>
> A5. Thanks for pointing out this issue! Please refer to Theorem 6.3 in the textbook by Polyanskiy and Wu, 2022. This reference will be incorporated into our revised paper.
>
> —Polyanskiy, Y. and Wu, Y., 2022. Information theory: From coding to learning.
>
> ---------
> **Q6. Could the authors elaborate on the group fairness achieved by their strategy vs the other baselines in the empirical analysis?**
>
> A6. (Please see response to Q1 as well). In all of our plots, the x-axis corresponds to a group fairness metric. For example, in Figure 2, the baseline models are represented by the red curves. The baseline model is a logistic regression classifier combined with the DispMistreatment fairness intervention [58] and FairProjection [2] on the left and right, respectively, both using zero imputation to handle missing values. Their fairness-accuracy trade-off is pareto-dominated by the proposed methods for preserving information about missing values. We observe a similar pattern, i.e. the proposed methods pareto-dominating the fairness-accuracy plot, across all experiments.

---

> > ### Comment · Reviewer_ZFNz · 2023-08-21
> > **Response**
> >
> > I thank the authors for their response! I looked at the discussion with other reviewers as well, and am satisfied with the revisions the authors will make. I will increase my score to a 7.

---

> > > ### Author Response · Authors · 2023-08-22
> > >
> > > Thanks so much for your response. We are glad to know that you are satisfied with our response. We will make sure to include the promised changes in the revised paper. Thank you again for the insightful and constructive comments you provided.

---

### Official Review · Reviewer_GNEv · 2023-07-12

**Soundness:** 2 fair
**Presentation:** 3 good
**Contribution:** 3 good
**Rating:** 7
**Confidence:** 3

**Summary:**

The paper works on the missing value issues in algorithmic fairness. Typical approaches tend to firstly impute the missing the data, then process for the classification task. However, the authors prove that the imputed data harms the group fairness as well as the averaged accuracy. To avoid losing missing pattern of the data to be imputed, the authors propose to modify the dataset to preserve the feature within the missing patterns, then continue with an off-the-shelf fairness-intervention algorithm to the modified dataset. Experiment results show that the proposed adaptive algorithm improves fairness and accuracy over impute-then-classify methods.

**Strengths:**

* Theoretical illustration on the conclusion that ``imputed data harms the group fairness as well as the averaged accuracy".

* The authors propose three methods for adapting linear fair classifiers to missing values:

     * **Method 1:** Adding missing indicator variables $\to$ this adaptive algorithm improves the accuracy of classifier under the same group fairness constraint than a classifier trained using impute-then-classify;

     * **Method 2:** Affinely adaptive classification;

     * **Method 3:** Missing pattern clustering;

* A general algorithm for  nonlinear classifiers.

* Experiments on various datasets demonstrate the effectiveness of the proposed methods.

**Weaknesses:**

* (1) Although the overall presentation is well-done, some parts could be much better if modified accordingly (please refer to **questions**).

* (2) Although the motivation of Theorem 1 is good, when I was going through the proof and assumptions made in Theorem 1, I feel like the assumptions are too strong, i.e., in this example, the conclusion should be based on:

  * (2a) The feature $X$ is of only one dimension; (it would be much better if it could be assumed as a two dimension, since missing values in one dimension feels like completely missing of a feature; while in real-world scenario, missing features are also like to be the case where part of information is missing in a feature $x=[x_1, x_2]$, i.e., $x_2$ is missing).

  * (2b) The construction of the probability distribution $P_{S, X, Y}$: given the attribute is $S=s$, it seems that the authors are requiring $Y=1$ won't appear in non-missing $X$ (as specified: $\text{Pr}(Y=1, X=0|S=s)\text{Pr}(Y=1, X=1|S=s)=0$), and $Y=1$ appears only for missing values (as specified:  $\text{Pr}(Y=1, X=\text{NA}|S=s)=\alpha_s, \text{Pr}(Y=1, X=\text{NA}|S=s)=0$).

Although the example itself is correct, it is hard to believe whether the conclusion will remain the same in more complex scenario.

**Questions:**

* (1) It would be much better if the saying of **missing values** could be explained at the beginning of the paper, since missing values may indicate many aspects, for example, missing labels $y$ of a sample $(x, y, z),$ or missing (hidden) attributes $z$, or missing instances $x$.

* (2) The notation w.r.t. accuracy $\mathbb{I}$ is given without any explanations.

* (3) Should there be any conditional independency between $\hat{X}$ and $Y$, in line 190?

**Limitations:**

Please refer to the section of **Weakness**.

---

> ### Author Rebuttal · Authors · 2023-08-09
>
> We thank the reviewer for the kind comments and the encouragement!
>
> ---------
> **Q1. Although the overall presentation is well-done, some parts could be much better if modified accordingly (please refer to questions).**
>
> A1. We appreciate your constructive feedback! Please find our detailed responses to your questions below, where we hope to adequately address all your concerns.
>
> ---------
>
> **Q2. Although the motivation of Theorem 1 is good, when I was going through the proof and assumptions made in Theorem 1, I feel like the assumptions are too strong.**
>
> A2. This is a great point! Regarding your (2a), Theorem 1 can indeed be generalized to a scenario where $X$ is made up of two variables $X_{obs}$ and $X_{ms}$. In this situation, $X_{obs}$ is always observed while $X_{ms}$ has missing values according to a certain probability. Under these setting, Theorem 1 remains valid; however, the mutual information is substituted by the conditional mutual information $I(M;Y|X_{obs})$. Regarding your (2b), our theorem relies on this assumption to maximize the dependency of the predicted label on the missing pattern. We could extend our results to a more general scenario where $Pr(Y=1,X=0|S=s)\neq 0$, but this might result in a looser upper bound in Theorem 1.
>
>
> ---------
> **Q3. It would be much better if the saying of missing values could be explained at the beginning of the paper, since missing values may indicate many aspects, for example, missing labels $y$ of a sample $(x,y,z)$ or missing (hidden) attributes $z$, or missing instances $x$.**
>
> A3. Thank you for the suggestion! In this paper, we focus on a specific setting where the input variables $x$ may have missing values. We will clarify this aspect in our abstract.
>
> ---------
> **Q4. The notation w.r.t. accuracy $\mathbb{I}$ is given without any explanations.**
>
> A4. Thanks for highlighting this issue. To clarify, $\mathbb{I}$ stands for the indicator function. It is defined as $\mathbb{I}(event) = 1$ if the event is true, otherwise, it is 0. We will add this definition at the beginning of Section 2.
>
> ---------
> **Q5. Should there be any conditional independence between $\hat{X}$ and $Y$ in line 190?**
>
> A5. You are indeed correct! The imputed variable $\hat{X}$ is derived by applying an imputation mechanism to $X$ without knowledge of $Y$ (since $Y$ is the predicted label), which yields the Markov chain: $Y–X–\hat{X}$, meaning that $\hat{X}$ and $Y$ are independent when conditioned on the observed value of $X$. In light of this, we use the data processing inequality, which allows us to arrive at the desired inequality: $I(Y; X) \geq I(Y; \hat{X})$.

---

> > ### Comment · Reviewer_GNEv · 2023-08-21
> >
> > Thanks for the responses, my concerns are well addressed. Hence, I increased my rating from 6 to 7.

---

> > > ### Author Response · Authors · 2023-08-22
> > >
> > > Thanks so much for your response. We are glad to learn that your concerns have been addressed. Thank you again for the insightful and constructive comments you provided.

---

### Official Review · Reviewer_oma2 · 2023-07-31

**Soundness:** 3 good
**Presentation:** 3 good
**Contribution:** 3 good
**Rating:** 6
**Confidence:** 4

**Summary:**

This work examines the impacts of missing values in data on fairness interventions, particularly in contrast to the commonly implemented "impute-then-classify" procedure for handling missing values. The authors present the following:
- investigation of how missing values impact algorithmic fairness in the context of three main modes of missing data (missing completely at random, missing at random, and missing not at random);
- a theorem capturing the performance gap between optimal solutions when employing a generic imputation mechanism vs. not when using equalized odds (information-theoretic result);
- methods for adapting fairness-intervention algorithms to missing data, both for linear and non-linear settings;
- empirical evaluation of their proposed methods.

Their findings suggest that fairness intervention strategies benefit from the preservation of information encoded in the missingness of data in terms of group fairness and accuracy.

**Strengths:**

- Clear motivation of the problem setting, both in the introduction and recapping in the conclusion.

- Overall, mathematical notation is quite clean and easy to follow.

- The authors provide an interesting information-theoretic performance gap result under a general imputation mechanism in the context of classification accuracy and equalized odds fairness constraint. This result implies that following imputation, then classification will never perform better (in terms of accuracy and group fairness) than using the information encoded in missing features, and moreover will result sub-optimal performance due to information loss.

- Multiple methods are presented to address missing values in the context of linear classification and one bagging-based method for nonlinear classification. These missing-value adaptation methods are flexible and can be used in conjunction with preexisting fairness-intervention algorithms (used in a black-box way).

- The authors provide meaningful discussion of challenges and limitations in using their methods (determining choice of fairness intervention, which the adaptation methods depend on; sensitive groups/attributes not being known beforehand). They also effectively demonstrate the value of this research direction in the context of algorithmic fairness.

- The authors provide implementation and hyperparameter details used in their experiments in the Appendix, supporting reproducibility (though this should be referenced in the main paper).

**Weaknesses:**

It's unclear what the trade-offs are between the three methods presented for linear classification. The figures comparing the performance between the methods against baselines are very hard to visually interpret, and there is insufficient discussion highlighting the performance differences between these and the baselines. It'd be helpful to further flesh out this section. It also makes it unclear what the value is in providing three methods for a more constrained and less practically-applicable setting given this presentation.

Furthermore, it is unclear how one would determine which of the three linear methods one should use - the authors note "we believe that the best adaptive algorithm is not universal, and one should select the adaptive algorithm based on the distribution of the data". What suggestions do you have for the reader in doing this?

Additional suggestions:
- In alignment with a question provided in the Questions section, proofs in the Appendix would benefit in some places from more rationale between steps - readers may not necessarily share your same mental model or background, and this can help cognitive overhead on the reader.

- Overall, the empirical results presented in the plots (Figures 1-3) are very hard to read and interpret, and are unfortunately quite inaccessible (font size, curve markers and overlap, overall size). The paper would benefit from making these more human-interpretable and by highlighting key takeaways (as stated above) and trade-offs.

- Nit: Please state upfront in Section 4 the settings you provide algorithms for! Based on the organization of the paper, a reader would have to be motivated to get to Section 5 to uncover that you provide methods for linear and nonlinear settings. :]

- Nit: please include references in the main paper to additional results/content in the Appendix throughout, i.e. proofs, additional experimental details, etc.

- Minor nit: please include the year in citation references.

**Questions:**

1. In the proof of Theorem 1, I didn't follow the transition between lines 539 and 540, nor where q came from. It'd be helpful to explicitly add a note for arriving at a < 1/3.

2. What implications does Theorem 1 have in a less trivial/higher dimensional setting? The multiclass classification setting?

3. Please refer to the Weaknesses/Suggestions section for additional questions around the linear adaptation methods, results, and trade-offs.

**Limitations:**

The authors provide meaningful discussion of challenges and limitations in using their methods, namely that the choice of fairness intervention is an important dependency and choice when utilizing the adaptation methods, along with the challenges introduced when sensitive groups/attributes are not known ahead of time and need to be inferred in an online manner.

---

> ### Author Rebuttal · Authors · 2023-08-07
>
> We thank the reviewer for the thoughtful comments and for appreciating the novelty and value of the work!
>
> ---
> **Q1. Trade-offs between the three methods for linear classifiers.**
>
> Please refer to our response for Q2 below.
>
> **Q2. It is unclear how one would determine which of the three linear methods one should use - the authors note "we believe that the best adaptive algorithm is not universal, and one should select the adaptive algorithm based on the distribution of the data". What suggestions do you have for the reader in doing this?**
>
> A2. We thank the reviewer for raising this point. We introduced three adaptive algorithms for linear classifiers since, across our experiments in the main text and the appendix, no single method consistently dominates fairness-accuracy performance. Our suggestion is for users to consider the choice of intervention as a hyperparameter and select based on a validation set. We briefly summarize the trade-offs between these methods in terms of complexity and performance [this discussion will be added to the updated paper]:
>
> - Adding missing value indicators [Sec 4.1] is the simplest intervention, which is arguably its main appeal. The advantage of this approach in linear models is that model weights assigned to missing-value indicators may enable users to interpret (to the extent that linear models are interpretable) how missingness is incorporated in classification. Adding missing indicators can achieve comparable performance to the other two methods (cf. Fig 2, 7), though tends to perform worse on average.
> - Affinely adaptive classification [Sec 4.2] is more flexible than adding missing value indicators, with the drawback of requiring additional parameters in the model (see lines 204-207). This method performs competitively with adding missing indicator values and can achieve higher accuracy (cf. Fig 2, 7). However, the user should weigh the benefit of a potential gain in accuracy given against the additional complexity cost.
> - Missing pattern clustering [Sec 4.3] can achieve the best fairness-accuracy performance if the missing value patterns can be clustered such that applying separate fairness interventions on each cluster is advantageous. Note that here the classifier is linear per cluster, but not as a whole. Figure 6 in the appendix displays experiments on synthetic data where missing pattern clustering achieves a far better accuracy-fairness operation point. Appendix E.1 explains why this performance is observed. This gain in performance was more muted in the experiments in Figure 1 and 2. Ultimately, missing pattern clustering should be preferred when sufficient data is available to cluster missing patterns.
>
> ---
> **Q3. Providing more rationale between proof steps.**
>
> A3. Absolutely, that's an excellent suggestion! We'll certainly expand on the proofs in the Appendix, providing a more detailed explanation as well as offering intuitive insight. For additional information, please refer to our responses to Q8.
>
> ---
> **Q4. Interpretability and accessibility of plots (Figures 1-3).**
>
> A4. Please see above for a discussion of the trade-offs between different proposed methods. We will provide updated figures and captions to improve accessibility and interpretability in the revised paper.
>
> ---
>
> **Q5-7. Nits and minor nits stated in Weaknesses.**
>
> A5-7. Thank you for raising these important clarification points – we will update the manuscript addressing all the nits and minor nits. We will add additional pointers to the appendix in the main text.
>
> ---
> **Q8. In the proof of Theorem 1, I didn't follow the transition between lines 539 and 540, nor where q came from. It'd be helpful to explicitly add a note for arriving at a < 1/3.**
>
> A8. Thank you for carefully reviewing the proof. We provide additional details below which will be incorporated into the revised paper.
> By our constructed data distribution $P_{S,X,Y}$ in lines 531-532, we know
> $$
> Pr(Y=1, S=0)
> = (Pr(Y=1, X=0 | S=0) + Pr(Y=1, X=1 | S=0) + Pr(Y=1, X=NA | S=0)) \cdot Pr(S=0)
> = \alpha_0 q_0
> $$
> where $q_0$ denotes $Pr(S=0)$.
> Similarly, we have
> $$
> Pr(Y=1, S=1) = \alpha_1 q_1,
> Pr(Y=0, S=0) = (1-\alpha_0) q_0,
> Pr(Y=0, S=1) = (1-\alpha_1) q_1.
> $$
> Now we have
> $$
> Pr(\hat{Y} = Y) = Pr(\hat{Y} = 1| Y=1, S=0) Pr(Y=1, S=0)  + Pr(\hat{Y} = 1| Y=1, S=1) Pr(Y=1, S=1) + Pr(\hat{Y} = 0 | Y=0, S=0) Pr(Y=0, S=0) + Pr(\hat{Y} = 0 | Y=0, S=1) Pr(Y=0, S=1)
> = (1-p_1) (\alpha_0 q_0 + \alpha_1 q_1) + \frac{p_0 + p_1}{2} \cdot ((1-\alpha_0) q_0 + (1-\alpha_1) q_1)
> $$
> where the last step uses the equations between lines 538 and 539, coupled with the equations we elaborated above. Lastly, note that $q_0 + q_1 = 1$. This leads us to the desired equation.
>
> Regarding $\alpha < 1/3$, we apply it in the final step of the equation from line 540 to 541. Given this condition, we have $1-\alpha > 0$ and $1-3\alpha > 0$. As a result, the objective function reaches its maximum when $p_0$ and $p_1$ are both equal to 1.
>
> ---
> **Q9. What implications does Theorem 1 have in a less trivial/higher dimensional setting? The multiclass classification setting?**
>
> A9. Yes, Theorem 1 can be extended to the multi-class setting. It can also be extended to the setting where $X$ is composed by two variables $X_{obs}$ and  $X_{ms}$ where  $X_{obs}$ is always observed and $X_{ms}$ has missing values according to a certain probability. In this case, the statement of Theorem 1 still holds while the mutual information is replaced by the conditional mutual information $I(M;Y|X_{obs})$.
>
> ---
> **Q10. Please refer to the Weaknesses/Suggestions section for additional questions around the linear adaptation methods, results, and trade-offs.**
>
> A10. We hope our response has addressed all of your comments. Please feel free to let us know if you have any additional comments that can help us further improve the paper.

---

### Author Rebuttal · Authors · 2023-08-09

We would like to thank all the reviewers for taking the time and effort to review our paper! We are delighted to receive positive feedback for the key components of the paper; in particular, that: our information-theoretic result characterizing the limitation of impute-then-classify provides novel and meaningful theoretical backing for our work (Reviewers oma2, SEin), our proposed algorithms are effective and interpretable (Reviewers GNEv, RyBS), and that our work sufficiently novel and addresses a significant problem in fair ML (Reviewer ZFNz). The thoughtful feedback we received is even more appreciated in light of reviewers having several papers to handle in a short period of time.

Below, we address the main points and questions raised by each reviewer and outline how we plan to update the paper accordingly; we will add the changes in the final version (both in the main text and appendix). Some common points shared by multiple reviewers are addressed below in this global response and are referred to accordingly in the responses to the individual reviewers.

We would also appreciate it if you could acknowledge that you read the response and, if your concerns are addressed, if you would kindly consider raising your review score. We also welcome any additional feedback or suggestions that could further strengthen our paper and would be glad to hear from the reviewers.

Thank you!

---

**Q1. Reviewers RyBS and SEin mentioned extending our approach to related settings, such as (a) missing sensitive attributes and/or labels and (b) multiaccuracy and multicalibration notions of fairness.**

A1. We thank the reviewers for raising these great points.

(a) We first note that while missing input features is a challenge with regard to handling missing patterns, missing sensitive attributes and/or labels is a challenge with regard to evaluating (group) fairness metrics and incorporating fairness constraints into model training. Hence, we cannot directly apply existing work on missing sensitive attributes or labels to deal with our missing-input-feature problem setting, nor vice versa. However, one can easily combine our approach with existing work on missing sensitive attributes/labels to provide a complete story that can deal with any possible missing features.

As an example, Yan et al. (2020) use a preprocessing method involving clustering and resampling to improve class balance when sensitive attributes are unknown, prior to training a classifier such as logistic regression. If there are missing values in non-sensitive attribute input features, we can apply our method by e.g. adding missing indicators after the preprocessing scheme, right before classifier training.

The use of sensitive attributes when adapting non-linear fair classifiers (Section 5) is more complex as the sensitive attribute is explicitly used when drawing subsamples from the dataset; we make this limitation explicit in lines 355-358.

(b) We recognize and appreciate the rigor in the definitions of multiaccuracy and multicalibration, as well as the growing literature surrounding it. In practice, any of the three methods in Section 4 can be used prior to applying a multiaccuracy/multicalibration fairness intervention since they do not depend on specific knowledge of the group attribute (as mentioned above) and only require adding more features/parameters (sections 4.1 and 4.2) or training separate fairness interventions on different part of the dataset based on missing value pattern (section 4.3). Adapting the method in Section 5 for multiaccuracy and multicalibration is in contrast non-trivial, since it is tailored to the group fairness setting where group attributes are in a discrete set. Consequently, it cannot be directly extended to the setting where groups are denoted by a finite-complexity (yet potentially uncountable) set of functions against which the error should be uncorrelated (multiaccuracy) or calibrated (multicalibration). Nevertheless, the current experiments in the paper support our main conclusion: not preserving missingess information (such as in the impute-then-classify approach) can hinder the performance of fairness interventions.

References:

Yan, S., Kao, H. T., & Ferrara, E. (2020, October). Fair class balancing: Enhancing model fairness without observing sensitive attributes.

---

### Decision · Program_Chairs · 2023-09-21

**Decision:**

Accept (poster)

**Comment:**

This study delves into the influence of missing data on the effectiveness of fairness interventions. The rationale behind this exploration stems from concerns that the widely employed "impute-then-classify" approach for handling missing values may hinder the process of ensuring fairness.
The research encompasses the following key aspects:
1. An investigation into how missing values impact algorithmic fairness across three primary modes of missing data (missing completely at random, missing at random, and missing not at random).
2. Development of a theorem that quantifies the performance gap between optimal solutions when using a generic imputation mechanism versus when fairness constraints, such as equalized odds, are applied (an information-theoretic result).
3. Methodological approaches for adapting fairness-intervention algorithms to accommodate missing data, applicable to both linear and non-linear scenarios. Companioning with empirical evaluations of the proposed methods, the authors demonstrate the practical insights into their efficacy.

The reviewers are unanimous in the paper’s contributions and the technical soundness. The authors are encouraged to incorporate the discussions during the rebuttal into their final version.